# Posterior Sampling for Reinforcement Learning on Graphs

**Arnaud Robert**                                                    *a.robert20@imperial.ac.uk*
*Department of Computing*
*Imperial College London*

**A. Aldo Faisal**                                                    *a.faisal@imperial.ac.uk*
*Department of Computing*
*Imperial College London*

**Ciara Pike-Burke**                                                  *c.pike-burke@imperial.ac.uk*
*Department of Mathematics*
*Imperial College London*

**Reviewed on OpenReview:** *https://openreview.net/forum?id=kd6CfmdPfX*

## Abstract

Many Markov Decision Processes (MDPs) exhibit structure in their state and action spaces that is not exploited. We consider the case where the structure can be modelled using a directed acyclic graph (DAG) composed of nodes and edges. In this case, each node has a state, and the state transition dynamics are influenced by the states and actions at its parent nodes. We propose an MDP framework, *Directed Acyclic Markov Decision Process* (DAMDP) that formalises this problem, and we develop algorithms to perform planning and learning. Crucially, DAMDPs retain many of the benefits of MDPs, as we can show that Dynamic Programming can find the optimal policy in known DAMDPs. We also demonstrate how to perform Reinforcement Learning in DAMDPs when the transition probabilities and the reward function are unknown. To this end, we derive a posterior sampling-based algorithm that is able to leverage the graph structure to boost learning efficiency. Moreover, we obtain a theoretical bound on the Bayesian regret for this algorithm, which directly shows the efficiency gain from considering the graph structure. We then conclude by empirically demonstrating that by harnessing the DAMDP, our algorithm outperforms traditional posterior sampling for Reinforcement Learning in both a maximum flow problem and a real-world wind farm optimisation task.

## 1 Introduction

Reinforcement Learning (RL) algorithms are typically used to solve Markov Decision Processes (MDPs), a general model for sequential decision-making under uncertainty in which the state of the environment only depends on what happened in the previous time step. There exist many online reinforcement learning approaches for finding an optimal policy in an unknown MDP (e.g. Watkins & Dayan (1992); Sutton et al. (1999); Auer et al. (2008); Azar et al. (2017); Strens (2000)). However, these methods tend to be quite general and do not make any assumptions beyond the MDP structure. This generality means that while these methods can be applied to most MDP problems, they will fail to exploit any additional structure present in the problem and, as such, can be sub-optimal in specific cases.

In this paper, we consider a special case of Markov Decision Processes in which the environment's dependency structure is encoded in a directed acyclic graph. In such a setting, the state space can be decomposed into the state at the nodes of the graph, and the reward and transitions depend on the connectivity of each node. In particular, the state at each node of the graph will depend on the state and action taken at all its parent nodes.

This structure is present in a lot of real-world settings. For example, in wind farm optimisation, e.g. Abkar et al. (2023), a grid of rotating wind turbines creates a stream of turbulence that impacts the yield of downstream wind turbines. The turbulence at downstream wind turbines depends on the orientation of the upstream wind turbines. Finding the optimal orientation of all wind turbines in the wind farm becomes an interesting control problem. Since the turbines' locations are known, the interaction pattern between all turbines composing the farm is also known. This interaction pattern can be encoded as a directed acyclic graph, where each node represents a wind turbine and each edge indicates a potential interaction. The graph structure we consider also covers any problem formulated as a maximum flow problem (i.e. network routing Mammeri (2019) or supply chain management Rolf et al. (2023)). Such problems are typically modelled as directed acyclic graphs where the state at each node depends on the action taken at the parent nodes, and we observe the outcome of the actions at the parent nodes before taking actions at the child nodes.

This paper will show that leveraging this latent graphical structure can significantly improve performance, both in theory and practice. We develop a Posterior Sampling algorithm that effectively exploits graphical structures. We empirically show that the proposed algorithm outperforms algorithms that ignore the latent structure. We also theoretically validate the improved efficiency of the proposed algorithm through an upper bound on the Bayesian regret. This analysis is achieved by a careful analysis of the proposed algorithm, which breaks down the combinatorial structure of the state and action spaces and reuses repeated patterns within the graph. Note that while there are some similarities between the DAMDPs that we consider and the Factored MDP (FMDP) framework (Boutilier et al., 2000), there are significant differences that necessitate further innovation. As elaborated in Section 7, the main difference is that FMDPs consider a time-homogeneous setting, where the state factorisation remains the same for every time step. On the other hand, we consider a time inhomogeneous setting where the factorisation structure might change over time; this allows us to capture more complex and realistic graph structures.

**Contribution:** To demonstrate the efficiency gains from leveraging a latent graphical structure, we start by formalizing the sub-class of MDPs we focus on and then show that when the transition function and rewards are known, a dynamic programming approach can be applied to find an optimal policy. When the rewards and transitions are unknown, we develop a posterior sampling algorithm and show an upper bound on its Bayesian regret. This upper bound demonstrates an improvement compared to methods that ignore the latent graphical structure, such as Auer et al. (2008); Osband et al. (2013). We also provide empirical validation of our method's performance gain, first on a maximum flow problem and then on a wind farm optimization problem. To summarize, this paper proposes, analyses and evaluates a novel posterior sampling algorithm specifically designed to exploit the graphical structure present in many real-world problems.

## 2 Background

Before we introduce our decision-making framework, it is helpful to recall the foundations of Markov Decision Processes and Directed Acyclic Graphs.

### 2.1 Fixed-Horizon Markov Decision Process

A fixed-horizon time-inhomogeneous MDP is a tuple $M = \langle \{\mathcal{S}_t\}_{t=1}^H, \{\mathcal{A}_t\}_{t=1}^H, \{R_t\}_{t=1}^H, \{P_t\}_{t=1}^H, H, \rho \rangle$. We consider finite time-dependent state spaces $\mathcal{S}_t$ and action spaces $\mathcal{A}_t$ for $t \in \{1, \ldots, H\}$. The mean reward function is defined by $R_t : \mathcal{S}_t \times \mathcal{A}_t \to [0, 1]$ for all $t \in [H]$. In state $s_t$, after performing action $a_t$, the agent observes a reward $R_t(s_t, a_t) + \eta_t$, where $\eta_t$ are independent identically distributed sub-Gaussian noise ($\sigma = 1$). The probability of reaching a specific state $s_{t+1} \in \mathcal{S}_{t+1}$ when action $a_t \in \mathcal{A}_t$ was performed in state $s_t \in \mathcal{S}_t$ is determined by $P_t(s_{t+1}|s_t, a_t)$. The agent interacts with the environment during episodes of length $H$, and in each episode, the initial state $s_1 \sim \rho$ is drawn from the initial state distribution. For the following $H$ time steps, $t \in \{1, \cdots, H\}$, the agent observes a state $s_t \in \mathcal{S}_t$ and decides to perform an action $a_t$; the result of this action is immediately observed as a new state $s_{t+1} \sim P_t(\cdot|s_t, a_t)$ and an immediate reward $R_t(s_t, a_t) + \eta_t$.

Reinforcement learning algorithms aim to find policies $\mu_t : \mathcal{S}_t \to \mathcal{A}_t$ for all $t \in \{1, \cdots, H\}$ that maximise the cumulative reward over an episode. We measure this in terms of the value function:

$$V_{\mu,t}^M(s_t) = \mathbb{E}\left[\sum_{h=t}^H R_h(s_h, a_h)\Big| a_h = \mu_h(s_h), s_{h+1} \sim P_h(\cdot|s_h, a_h), s_t = s\right], \tag{1}$$

where the expectation is taken the stochastic transition dynamics $s_{h+1} \sim P_h(\cdot|s_h, a_h)$. Often, we evaluate the quality of a policy $\mu$ in terms of the value function of the initial state $s_1$, $V_{\mu,1}^M(s_1)$, where the superscript $M$ indicates that we compute the value in the MDP $M$. The aim is to find an optimal policy $\mu^*$ within the set of Markov deterministic policies $\Pi$, which maximizes the value function,

$$\mu^* \in \arg\max_{\mu \in \Pi} V_{\mu,1}^M(s_1). \tag{2}$$

It is also helpful to define the Q-function as the expected reward the policy can obtain given that at time $t$ in state $s_t \in \mathcal{S}_t$ the action $a_t \in \mathcal{A}_t$ is executed, $Q_{\mu,t}(s_t, a_t) = r_t(s_t, a_t) + \sum_{s_{t+1} \in S_{t+1}} P_t(s_{t+1}|s_t, a_t)V_{\mu,t+1}^M(s_{t+1})$.

### 2.2 Directed Acyclic Graphs

**Definition 2.1** (Directed Acyclic Graph). A directed acyclic graph (DAG) $G = (V, E)$ is defined by a set of vertices $V$ and a set of edges $E$. Each edge has a direction associated with it. The graph is said to be acyclic if there is no node for which there exists a path (with one or more edges) that leads to itself.

For any DAG, there is a corresponding topological ordering of the nodes (Bang-Jensen & Gutin, 2008, Sec. 2.3.2). It assigns a layer to each node; the layer of a node is equal to the length of the longest path from the root to the current node. If the DAG does not have a root we can trivially add an artificial node that serves as a root. In this work, we consider this form of *layered directed acyclic graphs*, whose edges only connect nodes from adjacent layers.

**Definition 2.2** (Layered Directed Acyclic Graphs). A Layered, Directed Acyclic Graph (LDAG) is a DAG with a specific topology. It has a unique root node and a unique leaf node. All nodes are organised in layers, and all edges go from one layer to the next. Edges between nodes of the same layer are not allowed, as well as edges that connect nodes that are two or more layers away.

With a layered directed acyclic graph, it is clear that all edges go from a layer $l$ to a layer $l + 1$ with $l \in \{1, \cdots, H - 1\}$. Larger steps, backward steps, and transversal steps (within the same layer) are not allowed. Note that the assumption of LDAG does not lose any generality. We show in Appendix G.1 that for any DAG, we can construct an LDAG. Since the corresponding LDAG conserves the same connectivity patterns, such transformation has a limited impact on the complexity measure considered in this article. We illustrate and discuss this point further in Appendix G.

## 3 Directed Acyclic Markov Decision Process

We consider a particular sub-class of MDPs that can be rolled out on an LDAG. We define a Directed Acyclic Markov Decision Process (DAMDP) as a tuple $M_G = \langle \mathcal{X}, \{\mathcal{Y}_j\}_{j=1}^{U_r}, \{r^j\}_{j=1}^{U_r}, \{p^i\}_{i=1}^{U_\tau}, H, \{\rho_A^v\}_{v=1}^{n_1}, G\rangle$. An illustration of a simple DAMDP is given in the leftmost plot of Figure 1. Critically, in addition to the MDP components (described below), a DAMDP $M_G$ is structured by an LDAG, $G$. The nodes of an LDAG, $G$, can be organized in layers. A node belongs to a layer $l$ if the shortest path from a root node to the current node is of length $l$. The execution of an episode consists of a graph traversal from the first layer to the last layer of $G$. Each node has its own state, and the original MDP state can be recovered by combining all nodes' states of the current layer. The horizon $H$ of the DAMDP corresponds to the number of layers in $G = (V, E)$ since we sequentially observe the graph's layers. Edges $e \in E$ connect a node from a layer $l$ to a node in layer $l + 1$, for some $l \in \{1, \cdots H - 1\}$. At each time step, a layer of nodes of $G$ describes the current state of the DAMDP; that is, in layer $l$, the $n_l$ nodes that compose this layer describe the full state at time step $l$. For each layer $t$, we consider an arbitrary ordering of its nodes $\{1, \cdots, n_t\}$,

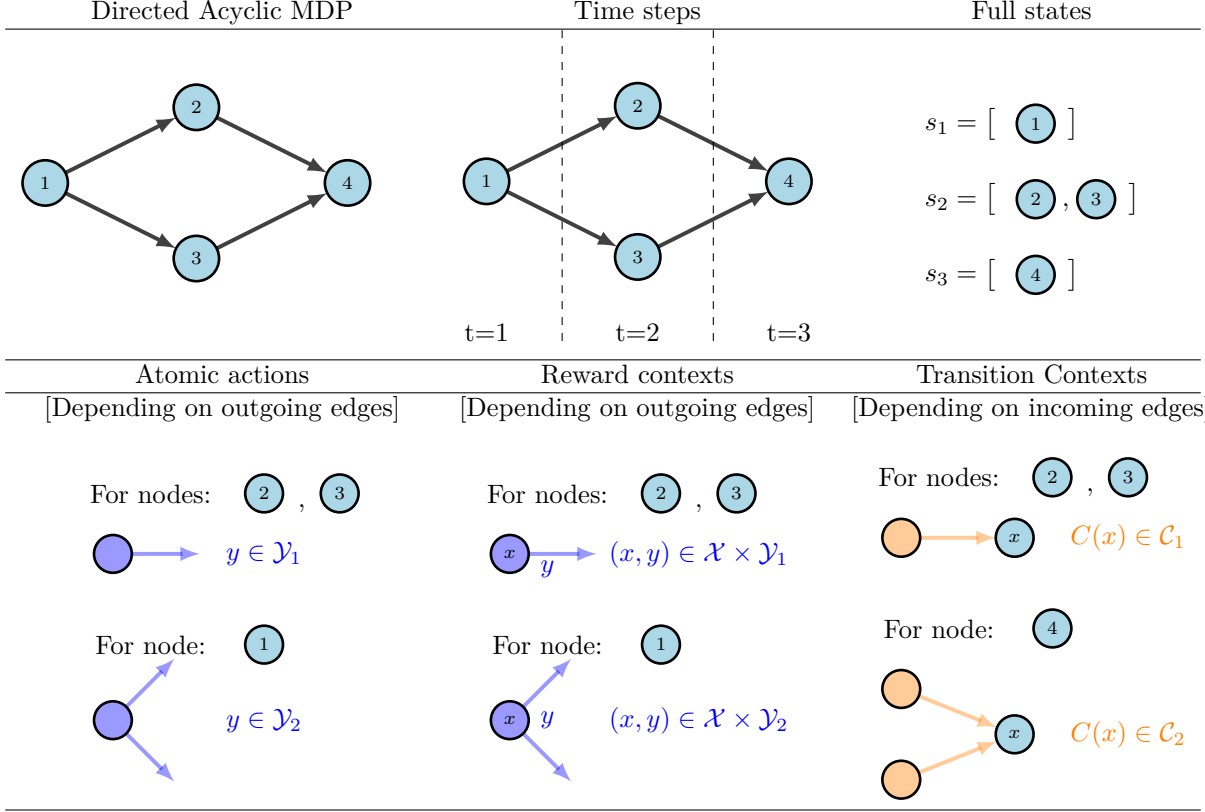

Figure 1: Illustration of the important components of DAMDPs. Inside the top leftmost box is an illustration of a simple DAMDP, $M_G$. The underlying graph has four nodes, which are visited within three time steps, $t = \{1, 2, 3\}$ as depicted on the second box of the top row. The last box of the top row illustrates the *full state* representation of $M_G$. At time step $t$, the *full state* $s_t$ consists of the concatenation of the observations at each node in layer $t$. The leftmost box of the bottom row shows the atomic action; in this particular example, the dimensionality of the atomic action space depends on the number of outgoing edges, i.e. the atomic action lies in $\mathcal{Y}_1$ if the node has a single outgoing edge, if the node has two outgoing edges the atomic action lies in $\mathcal{Y}_2$. The second box of the bottom row depicts the two atomic reward contexts that arise in $M_G$, which are composed of the *atomic state x* and the *atomic action y* of the current node. The last box of the bottom row shows the two atomic transition contexts that arise in $M_G$, which consists of the *atomic states* and *atomic actions* observed at the current node, $x$'s, parents (i.e. nodes from the previous time step). Consequently, the transition function of a node depends on the number of incoming edges.

which remains fixed for the entirety of the training. The edges of $G$ encode the dependence between different nodes. In a DAMDP, each node of $G$ has an *atomic state* $x$, where $x$ belongs to the *atomic state* space $\mathcal{X}$. In general, the *atomic* state space $\mathcal{X}$ is smaller than the original state space $\mathcal{S}_t$ for any time step $t \in [H]$. More precisely, the original state space $\mathcal{S}_t$ can be recovered by combining all the *atomic* state space of the $t^{th}$ layer nodes. Similarly, at each node, we can perform an *atomic action* $y \in \mathcal{Y}_j$ where $\{\mathcal{Y}_j\}_{j=1}^{U_r}$ denote the $U_r$ different *atomic action* spaces[1]. For a given time step $t \in [H]$, and a node $v \in [n_t]$ in layer $t$, which is in *atomic state* $x_t^v$, if the agent perform the *atomic action* $y_t^v \in \mathcal{Y}_j$, it receives a reward $r^j(x_t^v, y_t^v) + \eta_{t,v}$, where $r^j(x_t^v, y_t^v)$ is the mean atomic reward function associated with the $j^{th}$ atomic action space $\mathcal{Y}_j$, and $\eta_{t,v}$ is i.i.d. sub-gaussian noise. Here, and throughout, $v \in \{1, \cdots, n_t\}$ denotes the position of the current node in its layer, and we use the notation $x_t^v$ and $y_t^v$ to refer to the $v^{th}$ atomic state and action in the $t^{th}$ layer. If two nodes have the same number of parents and if their parents have the same *atomic action* spaces, then these two nodes obey the same dynamics and are said to belong to the same transition equivalence class $[\tau_i]$, with $i \in \{1, \cdots U_\tau\}$ and $U_\tau$ the number of equivalence classes. Formally, this implies that the dynamics of nodes that belong to the same equivalence class $[\tau_i]$ are governed by the same atomic transition distribution $p^i(\cdot|c)$, with $c$ the transition context, which consists of the atomic state and actions at all parent nodes. Hence, nodes with the same number of incoming edges have the same transition function. Concretely, let's consider the $v^{th}$ node of layer $t$ and assume that this node belongs to the equivalence class $[\tau_{i_{t,v}}]$, where we use the notation $i_{t,v}$ to denote the index of the equivalence class that node $v$ in layer $t$ belongs to. The atomic state at this node is drawn from the atomic transition function $x_t^v \sim p^{i_{t,v}}(\cdot|c_t^v)$. The transition context $c_t^v \in \mathfrak{C}_{i_{t,v}}$ consists of the concatenation of all parent's nodes *atomic states* and *atomic actions*. We denote by $\mathfrak{C}_{i_{t,v}}$ the space of all possible transition contexts, and note that this is the same for all nodes in the same equivalence class. We denote by $\mathcal{N}_t^v$ the set containing the parent nodes of the node $v$ in the $t^{th}$ layer, then the space of transition contexts for the $i^{th}$ equivalence class is, $\mathfrak{C}_i = \bigotimes_{v \in \mathcal{N}_t^v} \mathcal{X} \times \mathcal{Y}_{j_v}$, the Cartesian product of all possible configuration of the parent nodes *atomic states* and *atomic actions*. Finally, the initial *atomic state* components are sampled from initial atomic state distribution $x_1^v \sim \rho_A^v$ for all nodes of the initial layer of G, $v \in \{1, \cdots, n_1\}$.

We now illustrate the concepts introduced in this section with two practical examples. The key difference between the two examples lies in the construction of the atomic action spaces. The first example considers a single *atomic* action space $\mathcal{Y}$ for all nodes. While the second example considers that the *atomic* action space of a node depends on the number of outgoing edges, hence there are $U_r$ different atomic action spaces and $\{\mathcal{Y}_j\}_{j=1}^{U_r}$.

**Wind farm yield optimisation:** Optimisation of wind farm yield is an interesting control problem that can be modelled as a DAMDP. The extraction of kinetic energy and the interactions between the wind and the turbine's physical structure decrease the wind speed and increase the intensity of the turbulence in a wake region downstream of the wind turbine (Sedaghatizadeh et al., 2018). This wake effect impacts the yield of the downstream turbine. It is, however, possible to deflect this effect by modifying the yaw angle, the angle between the wind direction and the turbine head. While this can impact the turbine's immediate yield, it improves the wind conditions downstream and, consequently, the yield of the wind turbine in the original wake region. There is an optimal set of yaw angles that maximises the wind farm's total yield for a given farm layout and atmospheric conditions. When modelling this problem as a DAMDP, the graph $G = (V, E)$ encodes potential interactions between wind turbines. Each node, $v \in V$, represents a wind turbine in the farm. If an edge exists between node $v_1 \in V$ and node $v_2 \in V$, it indicates that the wake effect of wind turbine $v_1$ might impact wind turbine $v_2$. The *atomic* state space, $\mathcal{X}$, consists of the atmospheric conditions observed at a specific wind turbine. Similarly, the *atomic* action, $\mathcal{Y}$, characterises the yaw angle (i.e. the angle between the wind turbine and the upstream wind direction). Note that the atomic action space is the same for all nodes, reflecting the fact that all wind turbines can be configured with the same set of yaw angles. The *atomic* reward $r(x_t^v, y_t^v)$ observed at the node $v$ in layer $t$ is the yield of the corresponding wind turbine. In this example, the *atomic* reward context $(x_t^v, y_t^v)$ consists of the current node's *atomic* state $x_t^v \in \mathcal{X}$ and *atomic* actions $y_t^v \in \mathcal{Y}$. *Atomic* transitions functions, $p^i(x|c)$, determine the *atomic* state $x \in \mathcal{X}$

---

[1]In some applications, the *atomic action* description can be tightly linked to the number of outgoing edges at each node, hence it is natural to consider models where there are several different atomic action spaces.

based on the *atomic* transition context (i.e. the parent nodes *atomic* states and actions), $c \in \mathfrak{C}_i$, where $\mathfrak{C}_i = \bigotimes_{v \in \mathcal{N}} \mathcal{X} \times \mathcal{Y}$, with $\mathcal{N}$ the set of parent nodes.

**Maximimum leaky flow:** The maximum leaky flow problem Wayne (1999) is defined as a directed acyclic graph $G = (V, E)$, with a single root node, the source node $v_1 \in V$, and a single leaf node, called the sink node $v_N \in V$. Using edges to transport flow from one node to the next, the goal is to transport as much flow as possible from the source node to the sink node. Because $G$ is directed, it is only possible to transport flow in the edge's direction. Two parameters govern the environment: edges have a given capacity, $c > 0$, which denotes the maximum amount of flow that can be transported along that edge, and an unknown failure probability, $p \in [0, 1]$, which dictates how likely the edge is to fail to transport the flow assigned to it. This environment can also be modelled as a DAMDP. More specifically, the latent graph is $G$, and the atomic states space, $\mathcal{X}$, describe the amount of incoming flow in each node. The *atomic* action spaces, $\{\mathcal{Y}_j\}_{j=1}^{U_r}$, describe the proportion of incoming flow to assign to each outgoing edge. Note, in this particular problem instance, the *atomic* action space $\mathcal{Y}_j$ depends on the number of outgoing edges $j \in [U_r]$. Indeed, to specify how to distribute the flow among $n$ edges necessitates $n - 1$ entries. The *atomic* reward $r^j(x_t^v, y_t^v)$ is proportional to the amount of flow sent to the next layer by a node. The *atomic* reward context $(x_t^v, y_t^v)$ then consists of the node's *atomic* state $x_t^v \in \mathcal{X}$ and actions $y_t^v \in \mathcal{Y}_j$. Similarly, the transition function, $p^i(x|c)$, computes the distribution of flow at a node given its transition context. The transition context of a node, $c \in \mathfrak{C}_i$ consists of its parent's *atomic* states $x \in \mathcal{X}$ which represent the amount of flow received at each parent node and each parent's atomic action which is the proportion of flow they assign to the current node's incoming edges. Section 6.1) discusses this setting in more detail and includes visualization of several problem instances (see Fig. 6).

## 3.1 Relationship between DAMDPs and MDPs

A DAMDP $M_G = \langle \mathcal{X}, \{\mathcal{Y}_j\}_{j=1}^{U_r}, \{r^j\}_{j=1}^{U_r}, \{p^i\}_{i=1}^{U_\tau}, H, \rho_A, G \rangle$ is a special case of a finite horizon MDP, where the state and action spaces are time-dependent, but the atomic dynamics are stationary. While the number of nodes and the connection patterns might vary from one layer to the next, the atomic reward function and the atomic transition function behaviours remain unchanged. Below, we discuss the relationship between $M_G$ and the corresponding MDP $M$ constructed from the atomic components and the graph $G$.

**Time step and horizon:** The execution of a DAMDP episode is tightly linked to the graph $G$. Since the graph is layered, directed and acyclic, each layer $l$ corresponds to a time step $t$ in the corresponding MDP $M$. Hence, the depth $H$ of the graph corresponds to the MDP's horizon. After executing K episodes, the total number of time steps is $T = HK$.

**State space:** In any time step $t$, the *full state* $s_t \in \mathcal{S}_t$ is composed of the atomic state at each node in the graph's $t^{th}$ layer. Formally, in layer $t$ we observe $s_t = [x_t^1, \cdots, x_t^{n_t}]$, where $x_t^v$ is the observation collected in the $v^{th}$ node of layer $t$ (see Fig. 1, middle box) and $n_t$ denotes the number of nodes in layer $t$. For simplicity, we assume that each node has the same atomic state space, $\mathcal{X}$. Then the full state space at time step $t$ is the Cartesian product of the atomic state space of each node in layer $t$, $\mathcal{S}_t = \bigotimes_{v=1}^{n_t} \mathcal{X}$. We will use the term *full state* to refer to the concatenation of the observed values at each node of a layer, i.e. $s_t = [x_t^1, \cdots, x_t^{n_t}]$, which are the states observed in the corresponding MDP $M$. In contrast, an *atomic state* is the value observed at a specific node, i.e. $x_t^v$ for some $v \in \{1, \ldots, n_t\}$.

**Action space:** We similarly define an *atomic action* space, which is the set of actions that can be taken at a particular node. Note that all nodes do not necessarily have the same action space; $\{\mathcal{Y}_j\}_{j=1}^{U_r}$ represents the $U_r$ different *atomic action* spaces available. Let $y_t^v \in \mathcal{Y}_{j_{t,v}}$ be the *atomic action* of the $v^{th}$ node of layer $t$, and $\mathcal{Y}_{j_{t,v}}$ be the *atomic action* space associated with this specific node, with $j_{t,v} \in \{1, \cdots, U_r\}$ indicating which equivalence class node $v$ belongs to. The *full action* space at time step $t$ can be expressed as the Cartesian product of the *atomic action* space of each node composing layer $t$, $\mathcal{A}_t = \bigotimes_{v=1}^{n_t} \mathcal{Y}_{j_{t,v}}$. The *full action* at time step $t$, $a_t \in \mathcal{A}_t$, is the concatenation of the *atomic actions* selected at each node of layer $t$, $a_t = [y_t^1, \cdots, y_t^{n_t}]$. In Figure 1 (rightmost plot), we show examples where the dimension of the *atomic action* depends on the number of exiting edges.

**Policies:** Let $\mu = \{\mu_t\}_{t=1}^H$ denote a collection of time-dependent policies that maps a *full state* $s_t$ to a *full action*: $\mu_t : \mathcal{S}_t \to \mathcal{A}_t$ for all $t \in \{1, \cdots, H\}$. Since nodes in the same layer can have common successors, all *atomic actions* in a given layer must be jointly selected; ignoring these common dependencies and selecting *atomic actions* independently could lead to sub-optimal policies. Hence, the policies we consider still operate on the *full state* and *full action* spaces. Note that sometimes, we are interested in the action selected at a specific node. Let $a_t = \mu_t(s_t)$ be the *full action*, we denote the *atomic action* corresponding to the $v^{th}$ node of layer $t$ as $\mu_t(x_t^v|s_t) \in \mathcal{Y}_{j_{t,v}}$, to make it explicit that the policy requires knowledge about the *full state* in order to select *atomic actions*.

**Transition function:** We assume that the known latent graphical structure encodes conditional independence between nodes, with the state at a node depending only on the atomic state and actions taken at the parent nodes. We split nodes into equivalence classes $[\tau_i]$ based on their connectivity patterns, with all nodes in the same equivalence class having the same number of parents with the same atomic action spaces. The DAMDP is then governed by a set of atomic transition functions $\{p^i\}_{i=1}^{U_\tau}$. In particular, the probability of observing an atomic state $x_t^v \in \mathcal{X}$ at the $v^{th}$ node of layer $t$ is $p^{i_{t,v}}(x_t^v|c_t^v)$, where $c_t^v \in \mathfrak{C}_{i_{t,v}}$ denotes the *atomic state* and the *atomic actions* at all parent nodes and $i_{t,v}$ denotes the equivalence class of the node. Figure 1 (rightmost plot) shows all the transition contexts that arise in a simple DAMDP. The atomic transition function offers two immediate benefits. First, it reduces the number of dependent variables, focusing only on the subset of variables that influences the next atomic state. Second, it is possible that a layer contains more than a single node belonging to the equivalence class $[\tau_i]$; in that case, within a single step, the algorithm will observe several samples from the same atomic transition function, $p^i$, offering more opportunities to collect data and learn the true unknown dynamics. We sometimes want the transition context to reflect that a specific *full action* $a \in \mathcal{A}_{t-1}$ was executed in layer $t-1$. In such cases, we write the context of node $v \in \{1, \cdots, n_t\}$ in layer $t$ as $c_{t,a}^v$. The context of any node in layer $t$ can be constructed using the atomic description of $a$, $a = [y_{t-1}^1, \cdots, y_{t-1}^{n_{t-1}}]$. The probability of observing a new *full state* $s_{t+1}$, given that the *full action* $a_t$ has been executed in *full state* $s_t$ is given by $P_t(s_{t+1}|s_t, a_t) = \prod_{v=1}^{n_{t+1}} p^{i_{t,v}}(x_{t+1}^v|c_{t,a_t}^v)$, where $s_t = [x_t^1, \cdots, x_t^{n_t}]$, $s_{t+1} = [x_{t+1}^1, \cdots, x_{t+1}^{n_{t+1}}]$ and $c_{t,a}^v$ denotes the context of node $v$ in layer $t$ given that full action $a$ was selected. Similarly, if actions are selected by a given policy $\mu$, we make this relation explicit in our notation with $c_{t,\mu}^v$. Note that for the first transition, at $t=1$, we define the context to be the empty set, that is, $c_1^v = \emptyset$.

**Reward function:** We assume that the expected reward at a time step $t$ in the full MDP is the sum of the atomic expected reward observed at each node composing the $t^{th}$ layer, $R_t(s_t, a_t) = \sum_{v=1}^{n_t} r^{j_{t,v}}(x_t^v, y_t^v)$, where $x_t^v$ and $y_t^v$ are the *atomic states* and *atomic actions* at node $v$, $j_{t,v}$ denotes the action space available at the $v^{th}$ node of layer $t$, and $r^{j_{t,v}}(x, y)$ is the mean of the atomic reward distribution associated with atomic action set $\mathcal{Y}_{j_{t,v}}$.

### 3.2 Learning in a DAMDP

As depicted in Figure 1 (rightmost plot), $G$ can be partitioned into layers, each representing a time step, $t \in \{1, \ldots, H\}$, of the DAMDP. At each time step within an episode, $t$, the agent observes the state of each node within the $t^{th}$ layer of the graph. Note that the number of nodes, $n_t$, will change depending on the time step $t$. Hence, the dimensionality of the *full state* and the observation also depends on the time step $t$. After observing each node's *atomic state*, the agent chooses the *full action* it wishes to perform (recall that this corresponds to an atomic action being taken at each node). Again, the dimensionality of the *full action* will depend on the current time step. Once the *full action* is taken, the agent observes the reward obtained at each node in the current layer and the new *atomic states* in each node of the next layer. This process repeats until the algorithm goes through all the $H$ layers in the graph, which completes an episode.

**Value function:** Similar to the value function of a policy in an MDP, the value of a policy $\mu$ in a DAMDP consists of the expected reward accumulated at each node; when ambiguous, we use the subscript $M_G$ to indicate that this value function is computed under the DAMDP $M_G$, namely $\forall t \in \{1, \cdots, H\}$ and $\forall s \in \mathcal{S}_t$,

$$V_{\mu,t}^{M_G}(s) = \mathbb{E}\left[\sum_{h=t}^H \sum_{v=1}^{n_h} r^{j_{h,v}}(x_h^v, y_h^v) \,\bigg|\, y_h^v = \mu_h(x_h^v|s_h), s_{h+1} \sim P_h(\cdot|s_h, a_h), s_t = [x_t^1, \cdots, x_t^{n_t}]\right]. \tag{3}$$

Among all the Markov and deterministic policies that operate on the *full* state-action space $\mu \in \Pi$, we aim to find the optimal policy, $\mu^*$, i.e. the policy with the largest value function:

$$\mu^* = \arg\max_{\mu \in \Pi} V_{\mu,1}^{M_G}(s_1). \tag{4}$$

We consider a sequential learning setting where the learner interacts with the DAMPD over $K$ episodes. After each episode, the learner can use the information gathered to improve their policy in the next episode. To measure the learner's performance on this task, we define the regret, which compares, at each point in time, the performance of the current policy against the optimal policy $\mu^*$.

**Definition 3.1** (Regret). A reinforcement learning algorithm $\mathfrak{A}$ chooses for each episode $k \in \{1, \cdots, K\}$ the policy $\mu_k$ that interacts with the environment. The regret of algorithm $\mathfrak{A}$ over $K$ episodes is defined as,

$$Regret(K, \mathfrak{A}) = \sum_{k=1}^{K} \Delta_k, \tag{5}$$

where $\Delta_k$ denotes the difference of value function in the true DAMDP, $M_G^*$, between the optimal policy $\mu^*$ and the current policy $\mu_k$:

$$\Delta_k = \sum_{s \in \mathcal{S}_1} \rho(s) \left( V_{\mu^*,1}^{M_G^*}(s) - V_{\mu_k,1}^{M_G^*}(s) \right). \tag{6}$$

where $\rho(s) = \prod_{v=1}^{n_1} \rho_A^v(x_1^v)$ with $s = [x_1^1, \cdots, x_1^{n_1}]$.

Note that the regret is a stochastic quantity that depends on the random *atomic* dynamics of $M_G^*$ and the algorithm's sampling procedure. In the remainder of this paper, we consider the Bayesian regret, where we also take an expectation over all possible DAMDPs $M_G$ that are drawn from a prior over MDPs $f$,

$$\mathbb{E}[Regret(K, \mathfrak{A})] = \sum_{k=1}^{K} \mathbb{E}[\Delta_k].$$

Note that, in general, we use $t_k$ to denote the starting time of the $k^{th}$ episode, so $t_k = (k-1)H + 1$, and we denote with $t_k + i$ the $i^{th}$ time step of episode $k$.

## 4 Dynamic Programming on Graphs

We begin by considering how to learn an optimal policy in a known DAMDP. This will form an important building block of the reinforcement learning algorithm for DAMDPs presented in Section 5.

The planning algorithm 1 computes an optimal policy via backwards recursion. Dynamic programming algorithms rely on the self-consistency property of the Bellman operator. We show that there exists an atomic Bellman operator for any time step $t \in \{1, \cdots, H\}$, $\mathcal{T}_{\mu,t}^{M_G}$, that performs a one-step rollout according to the policy $\mu$, using only the atomic description of the DAMDP $M_G$,

$$(\mathcal{T}_{\mu,t}^{M_G} V_{\mu,t+1}^{M_G})(s_t) := \sum_{v=1}^{n_t} r^{j_t,v}(x_t^v, \mu_t(x_t^v|s_t)) + \sum_{s_{t+1} \in \mathcal{S}_{t+1}} \prod_{v=1}^{n_{t+1}} p^{i_t,v}(x_{t+1}^v | c_{t+1,\mu}^v) V_{\mu,t+1}^M(s_{t+1}), \tag{7}$$

for all $t \in \{1, \cdots, H\}$ and for all $s_t \in \mathcal{S}_t$. Here $s_t = [x_t^1, \cdots, x_t^{n_t}]$ and $c_{t,\mu}^v$ denotes the transition context of the $v^{th}$ node of layer $t$ when actions are selected by $\mu$.

The following lemma guarantees the self-consistency of the atomic Bellman operator in equation 7.

**Lemma 4.1** (Consistency of atomic Bellman operator). *For any DAMDP $M_G = \langle \mathcal{X}, \{\mathcal{Y}_j\}_{j=1}^{U_r}, \{r^j\}_{j=1}^{U_r}, \{p^i\}_{i=1}^{U_\tau}, H, \rho_A, G \rangle$ and policy $\mu$, the value function $V_{\mu,t}^{M_G}$ satisfies:*

$$V_{\mu,t}^{M_G}(s_t) = \mathcal{T}_{\mu,t}^{M_G} V_{\mu,t+1}^{M_G}(s_t), \tag{8}$$

*for $t \in \{1, \cdots, H-1\}$ and $s_t \in \mathcal{S}_t$, with $V_{\mu,H}^{M_G}(s_H) = \max_{a \in \mathcal{A}_H} R_H(s_H, a)$ for all $s_H \in \mathcal{S}_H$.*

---

**Algorithm 1** Planning on a DAMDP

---

**Input:** $\{\hat{p}^i\}_{i=1}^{U_\tau}, \{\hat{r}^j\}_{j=1}^{U_r}, G$

$u_H(s_H) = \max_{a \in \mathcal{A}_H} \sum_{v=1}^{n_H} \hat{r}^{j_{H,v}}(s_H^v, a^v) \quad \forall s_H \in \mathcal{S}_H$

**for** $t = H - 1, \cdots, 1$ **do**

   **for** $s_t \in \mathcal{S}_t$ **do**

      Evaluate $u_t(s_t)$ according to:

$$u_t(s_t) = \max_{a_t \in \mathcal{A}_t} \left\{ \sum_{v=1}^{n_t} \hat{r}^{j_{t,v}}(x_t^v, y_t^v) + \sum_{s_{t+1} \in \mathcal{S}_{t+1}} \prod_{v=1}^{n_{t+1}} \hat{p}^{i_{t,v}}(x_{t+1}^v | c_{t+1,a_t}^v) u_{t+1}(s_{t+1}) \right\},$$

      Set

$$\pi(s) = \arg\max_{a_t \in \mathcal{A}_t} \left\{ \sum_{v=1}^{n_t} \hat{r}^{j_{t,v}}(x_t^v, y_t^v) + \sum_{s_{t+1} \in \mathcal{S}_{t+1}} \prod_{v=1}^{n_{t+1}} \hat{p}^{i_{t,v}}(x_{t+1}^v | c_{t+1,a_t}^v) u_{t+1}(s_{t+1}) \right\}$$

      where $s_t = [x_t^1, \cdots, x_t^{n_t}]$ and $a_t = [y_t^1, \cdots, y_t^{n_t}]$.

   **end for**

**end for**

**Output:** $\pi$ and $u_1(s_1) \, \forall s_1 \in \mathcal{S}_1$

---

This property is directly inherited from the MDP assumption; for the sake of completeness, a proof for this lemma is given in Appendix B.

Before developing a reinforcement learning algorithm for DAMDPs in Section 5, we need to verify that there exists a planning algorithm that can compute the optimal policy for a given atomic transition distribution $p$ and atomic reward function $r$. We show in Theorem 4.2 that this can be done via dynamic programming as outlined in Algorithm 1. The algorithm leverages the one-step Bellman equation for DAMDP in equation 7. It starts by finding the optimal *full action* for each *full state* at the last layer, $H$. This is trivial as the last *full action* will not impact future rewards. For each time step $t \in \{H - 1, \cdots, 1\}$ and each *full state* $s_t \in \mathcal{S}_t$ it uses the one-step Bellman equation for DAMDP equation 7 to find the corresponding value $u_t(s)$ and the optimal *full action* $a^*$. The following theorem guarantees that Algorithm 1 retrieves the optimal policy $\mu^*$ and its associated value function $V_{\mu^*,1}^{M_G}$ for any known DAMDP $M_G$. The policy $\mu^*$ can be directly used to find the optimal *atomic actions*, since the optimal *full action*, $a_t^* = [y_t^{1,*}, \cdots, y_t^{n_t,*}]$, directly encodes the optimal *atomic action*, $y_t^{i,*}$ for each node $i \in \{1, \cdots, n_t\}$ of the $t^{th}$ layer of $G$.

**Theorem 4.2.** *If Algorithm 1 receives as input $\{p^i\}_{i=1}^{U_\tau}$ and $\{r_j\}_{j=1}^{U_r}$, the atomic reward and transition functions of a known DAMDP $M_G$, then it returns $u_t(s_t)$ and $\pi$, the optimal value function and policy, i.e.*

$$u_1(s_1) = V_{\mu^*,1}^{M_G}(s_1) \, \forall s_1 \in \mathcal{S}_1 \; and \; \pi = \mu^*. \tag{9}$$

The full proof is in Appendix C and follows the one presented in Puterman (2014, ch. 4).

*Remark* 4.3. Using the atomic description of the environment does not impact the computational complexity of the planning algorithm, which remains $\mathcal{O}(H|\mathcal{A}||\bar{\mathcal{S}}|^2)$, where $\bar{\mathcal{S}}$ denotes the largest *full state* space.

## 5 Posterior Sampling RL on Graphs

This section presents PSGRL, a posterior sampling-based algorithm that can successfully leverage the latent graphical structure present in DAMDPs. First, we describe the algorithm in Section 5.1. Second, we state the main result of this paper, which is an upper bound on the Bayesian regret suffered by PSGRL in Section 5.2. Finally, we conclude this section by providing intuition on how the atomic representation used by PSGRL leads to the observed efficiency gains in Section 5.3.

---

**Algorithm 2** Posterior sampling on graph MDPs (PSGRL)

---

**Input** Prior $f$ encoding $\mathcal{G}$
**for** episodes $k = 1, 2, \cdots$ **do**
    Sample $M_{G,k} \sim f(\cdot | D_{t_k})$
    Compute $\mu^{M_{G,k}}$ using Algorithm 1
    **for** time steps $t = 1, \cdots, H$ **do**
        Select and execute $a_t = \mu_k(s_t, t)$
        Observe $r_t = [r_t^1, \cdots, r_t^{n_t}]$ and $s_{t+1} = [x_{t+1}^1, \cdots, x_{t+1}^{n_{t+1}}]$
        Append $(s_t, a_t, r_t, s_{t+1})$ to $D_{t_{k+1}}$
    **end for**
**end for**

---

## 5.1 The algorithm

In Algorithm 2, we present our posterior sampling algorithm for DAMDPs, which leverages the rich underlying structure in the DAMDP setting. The algorithm proceeds in episodes. At the beginning of each episode $k$, the algorithm samples a DAMDP $M_{G,k} \sim f(\cdot | D_k)$, where $f(\cdot | D_k)$ represents the current belief about the true DAMDP $M_G^*$, given the observations collected up to episode $k$, $D_k$. Note that in the context of DAMDP, to sample $M_{G,k}$ the algorithm essentially samples a set of *atomic* transition functions $\{p^i\}_{i=1}^{U_\tau}$ and *atomic* reward functions $\{r^j\}_{j=1}^{U_r}$. The algorithm then computes the optimal policy $\mu^{M_{G,k}}$ on $M_{G,k}$ and runs the policy for an episode. The trajectory $(s_t, a_t, r_t, s_{t+1})_{t=1}^H$ is added to our dataset $D_k$. We then use $D_k$ to update our belief about the distribution of the true DAMDP $M_G^*$. We assume that each transition function is a categorical distribution governed by a set of parameters $\theta_{p^i} \in \mathbb{R}^X$, for all $i \in \{1, \cdots, U_\tau\}$. The expected reward function has a Gaussian distribution with mean $\mu_{r^j}$ and standard deviation $\sigma_{r^j}$, for $j \in \{1, \cdots, U_r\}$. The prior $f$ maintains a Dirichlet distribution over the space of parameters $\theta_{p^i}$ and a Normal-inverse gamma prior over the space of parameters $\mu_{r^j}$ and $\sigma_{r^j}$. This induces a prior distribution over DAMDPs.

## 5.2 Upper bound on the Bayesian regret

The regret suffered by PSGRL (Alg. 2) is bounded by the following theorem.

**Theorem 5.1.** *If $f$ is the distribution of $M_G^*$ then the regret suffered by PSGRL (Alg. 2) over $K$ episodes is upper bounded by,*

$$\mathbb{E}[Regret(K, PSGRL)] = \tilde{\mathcal{O}}\left( \sum_{j=1}^{U_r} H \sqrt{XY_j m_r^j K} + \sum_{i=1}^{U_\tau} H \sqrt{(X\bar{Y})^{d_i} m_\tau^i XK} \right) \tag{10}$$

*where $X = |\mathcal{X}|$ is the size of the atomic state space, $\bar{Y} = \max_{j \in \{1, \cdots, U_r\}} |\mathcal{Y}_j|$ is the size of the largest atomic action space. For nodes that belong to the $i^{th}$ equivalence class $[\tau_i]$, $d_i$ denotes the number of parent nodes, and $m_\tau^i$ denotes the number nodes in $G$ that share the same equivalence class $[\tau_i]$. Finally, $m_r^j$ is the number of nodes in $G$ with the same atomic action space $\mathcal{Y}_j$.*

The proof of Theorem 5.1 is given in Appendix D. It follows the proof in Osband et al. (2013), with the key difference being that the confidence sets are built around the estimates of the atomic transition and reward functions. This approach has two immediate benefits. First, we might be able to collect more than a single sample per time step. If we consider the case where we have $k$ nodes in the same layer, $t$, that belong to the same equivalence class, $[\tau_i]$, then at time step $t$, we collect $k$ samples from $p^i$. Second, the input space of the atomic reward and transition function can be significantly smaller than the input space in the original MDP (i.e. the transition and reward functions over the full state and action spaces). This leads to a significant gain in performance since the atomic transition and reward function are easier to estimate than their full state-action counterparts.

The bound presented in Theorem 5.1 is an improvement over the lower bound that can be obtained by running any RL algorithm in the full MDP and ignoring the latent graphical structure. Indeed, by Osband et al. (2013), the lower bound for an RL algorithm that ignores the graphical structure is:

$$\Omega\left(\sum_{h=1}^{H} \sqrt{S_t A_t K H}\right) = \Omega\left(\sum_{h=1}^{H} \sqrt{(X\underline{Y})^{n_t} K H}\right). \tag{11}$$

Where $\underline{Y} = \min_{j \in \{1, \cdots, U_r\}} Y_j$ is the size of the smallest action space in the DAMDP. The comparison of PSGRL upper bound in equation 10 and the lower bound in equation 11 suggests an improved efficiency for PSGRL compared to RL algorithms that ignore the latent graphical structure. In particular, note that the number of incoming edges for any node in a given layer $t$ is always bounded by the number of nodes at the previous layer $d_i \leq n_{t-1}$ where $i$ denotes the equivalence class of any node in layer $t$. In conclusion, the regret suffered by PSGRL, as described in equation 10, is smaller than the regret any RL algorithm that ignores the graphical structure $G$ could suffer as described in equation 11:

$$\sum_{j=1}^{U_r} H\sqrt{XY_j m_r^j K} + \sum_{i=1}^{U_\tau} H\sqrt{(X\bar{Y})^{d_i} m_\tau^i X K} \leq \sum_{h=1}^{H} \sqrt{(X\underline{Y})^{n_t} K H}.$$

### 5.3 Efficiency gains via compact confidence intervals

This section gives more insight into the sources of PSGRL's efficiency gains. Specifically, it highlights that we can leverage the atomic components of the DAMDP to construct more compact confidence intervals. This leads to the improved theoretical guarantees seen in Theorem 5.1.

The fact that both the transition and the reward functions can be written in terms of their atomic counterparts represents an opportunity to increase the algorithm's efficiency. Indeed, the size of the combinatorial spaces $\mathcal{S}_t$ and $\mathcal{A}_t$ can present a challenge when a learner is tasked to learn the *full* transition distribution $P$ and the *full* reward function $R$ directly. The inputs of the atomic transition functions and the atomic reward functions lie in spaces much smaller than their non-atomic counterparts, presenting an opportunity to speed up learning. The input space for the atomic transition function $i$ depends on the size of the context space, $|\mathfrak{C}_i|$, for $i \in \{1, \cdots U_\tau\}$, where $U_\tau$ is the number of equivalence classes. Critically, $|\mathfrak{C}_i|$ includes states and actions for only a subset of the nodes at the previous layer. This space is smaller than the *full state-action* space, which is the Cartesian product of the atomic state-action pairs of all nodes in the previous layer. Similarly, the atomic reward only depends on the atomic state $x \in \mathcal{X}$ and the atomic action $y \in \mathcal{Y}_j$, for $j \in \{1, \cdots, U_r\}$, which is also smaller than the *full state-action* space.

We can exploit the structure to achieve further efficiency gains by observing that the input of the atomic transition and reward function can be observed more than once per time step. If two nodes in the same layer have the same number of incoming edges, then they are learning the same atomic transition function. Similarly, if they have the same *atomic action* space $\mathcal{Y}_j$, then they are learning the same atomic reward function. This benefits the learner as they observe more samples corresponding to the atomic functions than their non-atomic counterparts.

PSGRL (i.e. Alg. 2) leverages the above two properties to achieve efficiency gains. To see this, consider the sampling procedure of PSGRL when it samples a candidate MDP $M_{G,k}$ from a set of plausible MDPs $\mathcal{M}_{G,k}$. The regret of the algorithm is tightly linked with the pace at which the confidence sets $\mathcal{M}_{G,k}$ concentrate around the true MDP $M_G^*$. The atomic representation of the transition and reward function significantly simplifies the construction of the confidence intervals, which leads to the observed efficiency gains. Indeed, the width of the confidence interval for the atomic transition function[2] depends on the square root of the size of the outcome space, $\sqrt{X} = \sqrt{|\mathcal{X}|}$. To see this, let $\hat{p}^i$ be an empirical estimate of the atomic transition function for nodes in the equivalence class $i \in [U_\tau]$ and $p^i$ be the true atomic transition function. The algorithm samples $\tilde{p}^i$ that are close to the current estimate $\hat{p}_k^i$, i.e. $||\tilde{p}^i(\cdot|c) - \hat{p}^i(\cdot|c)|| \leq \epsilon$ with probability greater than $1 - \delta$ for all $c \in \mathfrak{C}_i$ and for all $i \in [U_\tau]$. The width of the confidence interval can be determined by

---

[2] we focus on the transition function, although similar arguments can be made for the reward function.

the concentration inequality in (Weissman et al., 2003) to be $\epsilon \leq \sqrt{\frac{2}{n}\left(X \log\left(\frac{2}{\delta}\right)\right) + 2d_i \log\left(\frac{2U_\tau X \bar{Y} m_t^i k}{\delta}\right)}$.

This explicitly shows that the width of the confidence interval scales with the square root of the outcome space, $\sqrt{X}$. Algorithms that construct confidence intervals with the *full* representation of the transition function will have confidence intervals that scale with respect to $\sqrt{|\mathcal{S}|}$, which is much larger than $\sqrt{X}$, leading to larger regret.

Another advantage of the atomic representation that PSGRL exploits is that the number of observed atomic transitions is larger or equal to the number of observed full transitions. This is a direct consequence of the atomic transition being stationary and potentially appearing several times within a layer or in different layers. This potentially increases the number of observations for each atomic transition function more than once per step. PSGRL uses all the atomic observations to construct confidence sets. This can speed up learning as we see that the confidence intervals scale with $1/\sqrt{n}$ so as the number of observations increases, the confidence sets shrink. However, using all the atomic observations complicates the analysis as we need to account for the fact that we can observe a reward or a transition input more than once per time step, in contrast to the classic analysis of RL frameworks that assume that a single sample is collected at each time step.

## 6 Experiments

We now illustrate that the performance gains suggested by Theorem 5.1 manifest themselves empirically. We consider two experimental settings: the maximum leaky flow of a graph and wind farm yield optimization.

### 6.1 Maximum leaky flow of a graph

The maximum leaky flow problem (introduced in Sec. 3) is defined as a directed acyclic graph $G = (V, E)$, with a single root node, the source node $v_1 \in V$, and a single leaf node, called the sink node $v_N \in V$. Using the edges to transport flow from one node to the next, the goal is to transport as much flow as possible from the source node to the sink node. Because $G$ is directed, it is only possible to transport flow in the edge's direction. The environment is governed by two parameters: edges have a given capacity, $c > 0$, which denotes the maximum amount of flow that can be transported along that edge, and an unknown failure probability, $p \in [0, 1]$, which dictates how likely the edge is to fail to transport the flow assigned to it.

The above problem is similar to the graph maximum flow problem (Ford & Fulkerson, 1956). However, the leaky version of the problem is more challenging as the probability of failure is unknown to the learner. In the leaky version of the problem, the learner is tasked to jointly learn the unknown dynamics and the paths that maximise the expected flow.

We consider two algorithms for this problem. The proposed PSGRL, which has access to the graphical structure and an extension of the PSRL algorithm (Osband et al., 2013) that works in time-inhomogeneous full MDP settings, which does not know the graphical structure.

**PSGRL representation:** At each time step $t$, we observe the set of *atomic states* $\{x_t^v\}_{v=1}^{n_t}$, which represent the amount of flow into node $v$ of layer $t$. For each node, the agent decides how to distribute the current flow along its edges. For each node, the *atomic action* space consists of the possible allocation of flow along its edges; the action space is discrete and represents a possible fraction of the total flow. The atomic reward is then the amount of flow a specific node sends to the next layer of the graph.

**PSRL representation:** The *full state* representation at time step $t$, $s_t$ is a vector representing the amount of flow in each node in layer $t$. The *full action* representation is the flow assignment for all edges connecting nodes from layer $t$ to layer $t + 1$. The reward at time step $t$ is the total amount of flow the agent managed to move from layer $t$ to layer $t + 1$.

Figure 2 compares the performances of PSGRL and PSRL on the leaky maximum flow problem. The first row of Figure 2 depicts the DAMDP considered, while the second row shows the cumulated regret incurred by PSGRL (in blue) and PSRL (in red). The uncertainty estimates around the curve are obtained by running

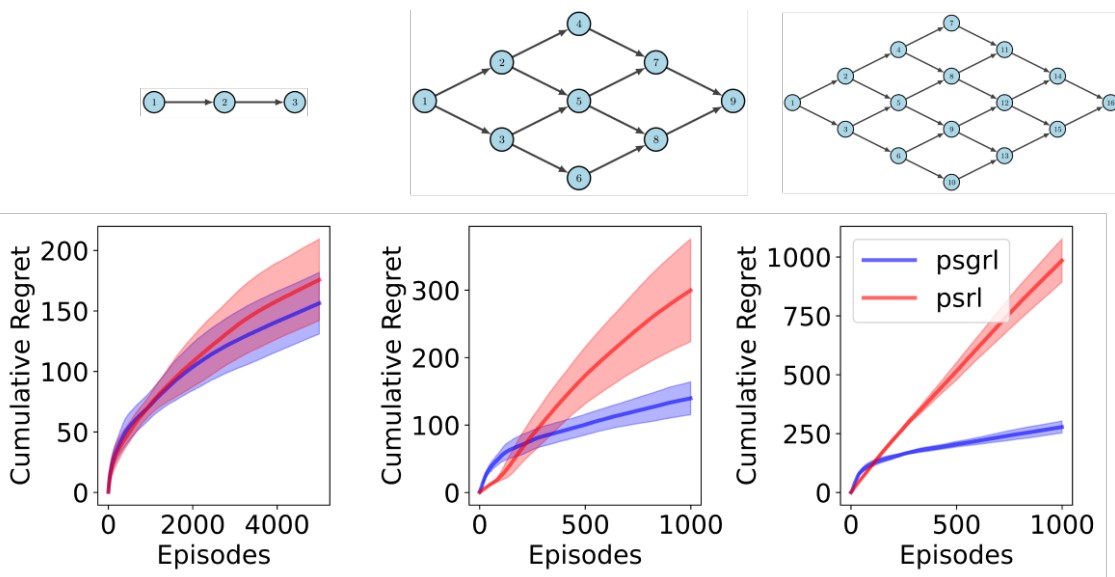

Figure 2: The first row depicts the graph that governs the maximum leaky flow problem instance. The second row shows the learning curve for both algorithms considered. PSRL ignores the latent graphical structure, and PSGRL leverages the graphical structure. The left-most plot shows the performance obtained on a simple chain graph. As expected, the performance for both algorithms is similar. Looking at the remaining plots, where we consider larger diamond-shaped graphs, it is clear that as we increase the complexity of the graph, the benefit of PSGRL becomes evident. To monitor the evolution of the regret, we ran ten different seeds; the solid line represents the mean regret while the shaded area covers ± one standard deviation.

the experiment with ten seeds. We present in Appendix H (see Figure 6) another series of experiments that highlights similar performance results on a different family of graphs.

The leftmost DAMDP is a simple chain graph. For this specific DAMDP instance, the *atomic* representation and the *full state* representation are equivalent. This explains the similar performance of PSGRL and PSRL. In contrast, considering more complex DAMDP instances, the benefit of leveraging the latent graphical structure becomes more apparent. Breaking down the combinatorial nature of the *full state* space and *full action* space using the DAG structure is beneficial. This is even more obvious on the rightmost plot, which shows the most complex leaky maximum flow instance considered.

*Remark* 6.1. Even for a simple example like the leaky maximum flow problem, it is interesting to analyse how the Bayesian regret bound for PSGRL equation 10 compares to the RL lower bound equation 11. The diamond-shaped graph with 16 nodes in the rightmost part of Figure 2 implies that the DAMDP is parameterized by the following quantities. The number of distinct atomic action spaces is $U_r = 2$ as some nodes can distribute their flow in a single edge while others can distribute it in two different edges. We observe that there are 6 nodes with a single outgoing edge, hence $m_r^1 = 6$ and 9 nodes with two outgoing edges, $m_r^2 = 9$. The number of transition equivalence classes is $U_\tau = 2$ as some nodes have two incoming edges while others have a single incoming edge. We observe that there are 6 nodes with a single incoming edge (i.e. $m_\tau^1 = 6$) and 9 with two incoming edges (i.e. $m_\tau^2 = 9$). We also note that the maximum number of nodes in a layer is $N_{max} = 4$, and the largest number of edges between two adjacent layers is $M_{max} = 6$. In this specific context, the regret upper bound of PSGRL can be expressed as follows:

$$\mathbb{E}[Regret(K, PSGRL)] = \tilde{\mathcal{O}}\bigg( \sum_{j=1}^{U_r} H\sqrt{2XY_j m_r^j K} + \sum_{i=1}^{U_\tau} H\sqrt{2(X\bar{Y})^{d_i} m_\tau^i XK} \bigg) \tag{12}$$

$$= \tilde{\mathcal{O}}\left( H\sqrt{12XY_1K} + H\sqrt{18XY_2K} + H\sqrt{12X^2\bar{Y}K} + H\sqrt{19X^3\bar{Y}^2K} \right). \tag{13}$$

In orange, we have the regret associated with learning the transition function, and in blue, the regret associated with learning the reward function. Similarly, using the characteristics of this specific problem

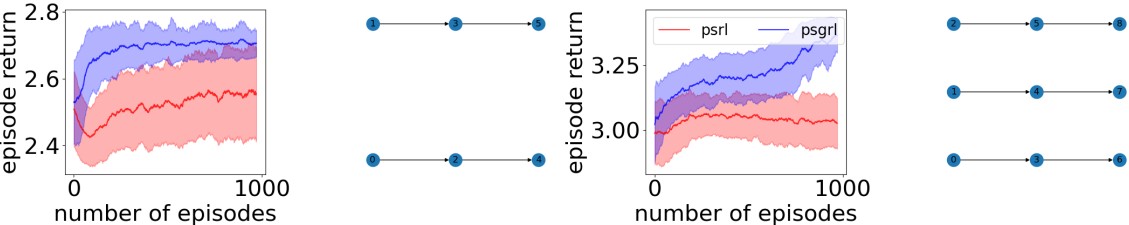

Figure 3: The two leftmost plots summarize the results obtained on a wind farm task with six wind turbines. The first plot shows the performance of PSGRL and PSRL, while the second plot illustrates the farm layout (each node represents a wind turbine, and each edge contains the wake effect). The two rightmost plots show the same experiment for a larger farm with nine wind turbines.

instance for any algorithm that do not leverage the latent graphical structure, such as PSRL, we obtain the following lower bound:

$$\mathbb{E}[Regret(K, PSRL)] = \Omega\left(\sum_{h=1}^{H} \sqrt{S_t A_t T}\right) = \Omega\left(\sum_{h=1}^{H} \sqrt{(X\underline{Y})^{n_t} T}\right) \tag{14}$$

$$= \Omega\big(2\sqrt{(X\underline{Y})^1 T} + 2\sqrt{(X\underline{Y})^2 T} + 2\sqrt{(X\underline{Y})^3 T} + \sqrt{(X\underline{Y})^4 T}\big), \tag{15}$$

where $\underline{Y}$ denotes the smallest atomic action space.

In this simple example, the efficiency gain provided by PSGRL will be significant as long as $\sqrt{X^3 \bar{Y}^2} < \sqrt{(X\underline{Y})^4}$ which is in general true, unless the discrepancy between the largest and smallest action space becomes too large, note that in the diamond graph considered we expect to see an efficiency gain as $\underline{Y} = Y$ and $\bar{Y} = Y^2$. Additionally, this efficiency gain is reinforced as we increase the size of the graph. Considering that the number of nodes increases while the diamond-like architecture on Figure 2 is preserved, PSGRL's regret equation 12 scales linearly with the number of nodes, through $m_\tau^i$ and $m_r^j$. On the other hand, PSRL's lower bound on the regret scales exponentially with the number of nodes through $n_t$, the number of nodes at each layer of the graph.

## 6.2 Wind farm yield optimisation

One real-world example with a known latent graphical structure is the case of wind farm optimization. To maximise the yield of a wind farm given a specific atmospheric condition (wind direction and speed), it is important to consider the interaction between wind turbines. Wind turbines generate a stream of turbulent wind that might impact the yield of a downstream wind turbine. This phenomenon is known as the wake effect. The wake of a wind turbine can be deflected if the upstream wind turbine slightly rotates. This introduces an interesting control problem as for a single wind turbine to maximize its yield, it has to face the wind, but as soon as the yield of the entire farm is concerned, the optimal set of angles might include wind turbines not facing the wind in order to maximize the yield of other wind turbines. In what follows, we constructed a simple environment that models some of the dynamics of a wind farm. To model the wake effect, we use FLORIS, a wind farm simulation software (Annoni et al., 2018) [3]. To simplify the setting, we discretize the *atomic action* $\mathcal{Y} = \{30°, 0°, -30°\}$ which are the possible angles (with respect to the wind direction) for each wind turbine. The *atomic state* encodes the wind speed observed at each wind turbine. We also discretize the state and consider all increments of $0.1m/s$ from $6m/s$ to $10m/s$. Similarly to the maximum flow experiment, the *full state* and *full action* at time step $t$ are obtained, respectively, by concatenating the *atomic state* of each node in layer $t$ and the *atomic action* of each node in layer $t$. The performance of PSGRL on the resulting DAMDP and PSRL on the corresponding MDP is shown in

---

[3]The code for the FLORIS simulator is available at the following address: https://github.com/NREL/floris

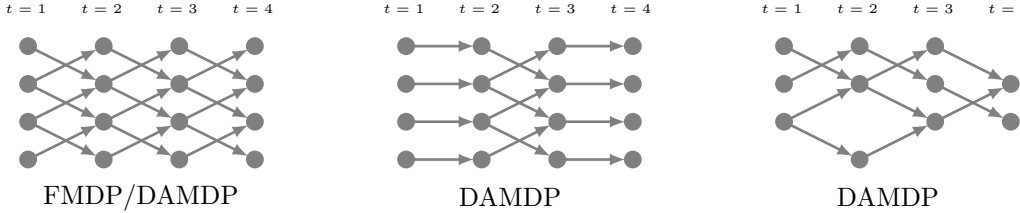

Figure 4: This figure shows three examples of DAMDPs; the leftmost one has a constant connectivity pattern. Each layer has the same number of nodes, and the connectivity pattern from one layer to the next remains the same. MDPs that exhibit such a structure could be modelled by an FMDP or a DAMDP. On the contrary, the middle and the rightmost graphs could only be modelled by DAMDPs. The example depicted in the middle graph shows interactions that vary from one time step to the next, which cannot be modelled by an FMDP. The rightmost graph shows a graph where the number of nodes varies from time step to time step, which also cannot be modelled as an FMDP

Figure 3, where each plot considers wind farms with a grid layout with a varying number of wind turbines. We consider a grid-like layout where we explicitly encode the impacts between a wind turbine and its closest upstream wind turbine. While the plots on the left of Figure 3 show a wind farm composed of six wind turbines, the right side of the figure shows an experiment with nine wind turbines. Again, it can be seen that the benefits of running PSGRL on the DAMDP are significant compared to the performance of PSRL on the corresponding MDP because of the re-usability of the atomic reward and transition function.

## 7 Related work

The DAMDP model considered in this paper is related to the factored MDP (FMDP) framework. While the first works on FMDP considered only a factorisation of the state space (Kearns & Koller, 1999; Boutilier et al., 2000; Guestrin et al., 2003), the FMDP framework was later extended to consider factorisation over the state-action space (Osband & Van Roy, 2014; Chen et al., 2020) - which in spirit is more similar to DAMDP. Indeed, FMDPs and DAMDPs share similar assumptions; they both consider the factorisation of the state and action spaces, the additivity of the reward function and the factorisation of the transition function. However, an important difference between the two frameworks is that FMDPs were initially proposed in a time-homogeneous setting, while DAMDPs allow the structure to vary as dictated by the graph and its connectivity.

In particular, DAMDP models a larger class of problems, as depicted in Figure 4; the encoding of the entire episode in a graph allows the representation of richer dynamics. The leftmost plot of Figure 4 shows a graph that exhibits the same connection patterns across all layers; such a problem could be modelled with FMDPs or DAMDPs. The remaining examples in the middle and on the right side of Figure 4 represent problems that cannot be directly modelled with FMDPs. The example in the middle of Figure 4 shows a scenario where the connection dynamics change from one layer to the next, while the rightmost example shows a graph with a varying number of nodes and edges in each layer. Since DAMDPs are rolled out on an arbitrary directed acyclic graph, we can model each of the three graphs in Figure 4 with a DAMDP. On the subset of DAMDP problems that can be modelled as FMDP, the regret obtained for the algorithm proposed in Osband & Van Roy (2014) is equivalent to the regret of our algorithm, PSGRL.

In general, DAMDPs, a naive extension of the regret proposed Osband & Van Roy (2014) to the time inhomogeneous setting, is worse than the result obtained in Theorem 5.1. A naive extension of the FMDP framework to the time inhomogeneous setting, where the factored dynamics are encoded in graph $G$, would produce a regret bound driven by the number of nodes in the graph $G$. To be more specific, using our notation, the regret paid by the posterior sampling-based algorithm proposed in Osband & Van Roy (2014) to estimate the transition distribution of an FMDP is $\sum_{v=1}^{|V|} \mathcal{O}(H(X\bar{Y})^{d_v} + \sqrt{(X\bar{Y})^{d_v} XT})$, where $|V|$ represents the number of nodes in $G$ and $d_v$ is the number of incoming edges in node $v$. For the case of DAMDPs, the bound we propose in Theorem 5.1 is tighter as in equation 10 the size of the transition context $(X\bar{Y})^{d_i}$ is

under the square root and, in general, $\sum_{v=1}^{|V|} (XY)^{d_v} \geq \sum_{i=1}^{U_\tau} \sqrt{(X\bar{Y})^{d_i}}$, as the first sum is over all nodes in the graph while the second is only over the $U_\tau$ equivalence classes. This difference becomes more evident if we consider increasing the number of nodes $|V|$ while preserving a relatively small number of equivalence classes $U_\tau$. For example, in the case of the diamond-shaped graph (e.g. Fig. 2), the graph's architecture remains constant, and the only term that grows is $m_t$, the largest number of nodes in a single layer that has the same transition context.

In this work, we have assumed that the graphical structure is known. An interesting direction for future work would consist of learning the latent graphical structure. Structure discovery has been studied in the case of time-homogeneous FMDP Degris et al. (2006); Strehl et al. (2007); Mutti et al. (2023), but these methods have not been extended to the time-inhomogeneous setting. Approaches like the one considered in Mutti et al. (2023), where the proposed algorithm, $C - PSRL$, can learn the factored representation, could be extended to the time-inhomogeneous setting. Such an extension is not trivial as it would require adding another layer of inference in our proposed algorithm and treating the latent graph $G$ as a random variable.

## 8 Conclusion

In this paper, we have formalised a new subclass of MDPs that have an additional graphical latent structure, DAMDPs. We then presented a posterior sampling-based algorithm, PSGRL, that exploits this graphical structure to efficiently learn a policy. We finally showed that PSGRL outperforms PSRL both in theory and in practice when this graphical structure is present.

Although the structure of DAMDPs may appear restrictive, we argue that it does appear in many practical settings, including the maximum flow of a graph and wind farm yield optimisation setting discussed in this paper. However, an immediate future step could be to extend our model. First, we could add edge attributes; for example, if we revisit the wind farm optimisation example presented in Section 6.2. We could imagine that the potential effect two wind turbines have on each other might depend on the relative distance between the two; adding additional attributes to the edge would allow us to consider a richer set of transition dynamics and to model real-world problems more realistically. Such an increase in complexity will impact the upper bound on the regret but would offer more flexibility in the modelling. Second, to accommodate a wider range of real-world problems, the proposed framework could be extended to allow for continuous state and action. Quantifying the benefit of the graphical factorisation in such a context remains of great interest. We leave these endeavours for future work as, for the time being, our graph-based approach renders a potentially large subset of real-world MDPs amenable to more efficient training. This phenomenon is even more evident when we consider that the graph computations are inherently parallelisable.

### Acknowledgments

AR was supported by an EPSRC CASE studentship supported by Shell, and AAF was supported by a UKRI Turing AI Fellowship (EP/V025449/1).

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

## A    Notation

| Symbol | Description |
|--------|-------------|
| $H$ | The episode's horizon. |
| $G$ | The additional structure encoded in a LDAG. |
| $\mathcal{X}$ | The atomic state space. |
| $\mathcal{S}$ | The full state space. |
| $\mathcal{Y}$ | The atomic action space. |
| $\mathcal{A}$ | The full action space. |
| $U_r$ | The number of different action spaces. |
| $r^j$ | An atomic reward function with $j \in [U_r]$. |
| $R_t$ | The full MDP's mean reward function at time ste $t \in [H]$. |
| $U_\tau$ | The number of nodes equivalence classes. |
| $p^i$ | The atomic transition function for nodes in equivalence class $i \in [U_\tau]$. |
| $P_t$ | The full MDP's transition function at time $t \in [H]$. |
| $\rho_A^v$ | The atomic initial distribution for node $v$ in the first layer of $G$. |
| $\rho$ | The full MDP's initial state distribution. |

## B    Proof of the consistency of the atomic Bellman operator

**Lemma B.1** (Consistency of the atomic Bellman operator). *For any DAMDP $M_G = \langle \mathcal{X}, \{\mathcal{Y}_j\}_{j=1}^{U_r}, \{r^j\}_{j=1}^{U_r}, \{p^i\}_{i=1}^{U_\tau}, H, \{\rho_A^v\}_{v=1}^{n_1}, G \rangle$ and policy $\mu$, the value function $V_\mu^{M_G}$ satisfies:*

$$V_{\mu,t}^{M_G}(s_t) = \mathcal{T}_{\mu,t}^{M_G} V_{\mu,t+1}^{M_G}(s_t), \tag{16}$$

*for $t \in \{1, \cdots, H-1\}$ and $s_t \in \mathcal{S}_t$, with $V_{\mu,H}^{M_G}(s_H) = \max_{a \in \mathcal{A}_H} R_H(s_H, a)$ for all $s_H \in \mathcal{S}_H$.*

*Proof.*

$$
\begin{aligned}
\mathcal{T}_{\mu,t}^{M_G} V_{\mu,t+1}^{M_G}(s_t) &= \sum_{v=1}^{n_t} r^{j_{t,v}}(x_t^v, \mu(x_t^v|s_t)) + \sum_{s_{t+1} \in \mathcal{S}_{t+1}} \prod_{v=1}^{n_{t+1}} p^{i_{t,v}}(x_{t+1}^v | c_{t+1,\mu}^v) V_{\mu,t+1}^{M_G}(s_{t+1}) \\
&= \sum_{v=1}^{n_t} r^{j_{t,v}}(x_t^v, \mu(x_t^v|s_t)) + \sum_{s_{t+1} \in \mathcal{S}_{t+1}} \prod_{v=1}^{n_{t+1}} \left( p^{i_{t,v}}(x_{t+1}^v | c_{t+1,\mu}^v) \mathbb{E}\left[ \sum_{h=t+1}^{H} \sum_{v=1}^{n_h} r^{j_{t+1,v}}(x_{t+1}^v, \mu(x_{t+1}^v|s_{t+1})) \right] \right) \\
&= \mathbb{E}\left[ \sum_{h=t}^{H} \sum_{v=1}^{n_h} r^{j_{t,v}}(x_t^v, \mu(x_t^v|s_t)) \right] \\
&= V_{\mu,t}^{M_G}(s_t)
\end{aligned}
$$

To obtain the first equality, we used the definition of the Bellman operator, and to obtain the second equality, we used the definition of $V_{\mu,t+1}^{M_G}$. Observing that we consider all possible next state $s_{t+1}$ weighted by its probability of appearing, we can directly include this computation into the expectation leading to the third equality. □

## C    Proof of convergence of Algorithm 1

**Theorem C.1.** *If Algorithm 1 receives as input $p$ and $r$, the atomic reward and transition functions of a known DAMDP $M_G$, then it returns $u_1(s_1)$ and $\pi$, the optimal value function and policy, i.e.*

$$u_1(s_1) = V_{\mu^*,1}^{M_G}(s_1) \text{ and } \pi = \mu^*. \tag{17}$$

*Proof.* The proof follows the one presented in Puterman (2014, ch. 4) and consists of two parts.

**Part i:** Show that the learned value function $u_t(s_t) \geq V_{\pi,t}^{M_G}(s_t)$ for all $\pi \in \Pi$ and for all $(t, s_t), \in \{1, \cdots, H\} \times \mathcal{S}_t$, where $\Pi$ is the class of all deterministic and Markov policies.

**Part ii:** Show that the algorithm's output satsifies $u_t(s_t) = V_{\pi^*,t}^{M_G}(s_t)$ for all $t$ and all $s \in \mathcal{S}_t$.

**Proof of i:** Let $\pi \in \Pi$ be any policy in the class of deterministic Markov policies. Since at the last time step $H$ the goal is only to maximise the immediate reward, the suggested result holds at time step $H$, $u_H(s_H) \geq V_{\pi,H}^{M_G}(s_H)$, for all $s_H \in \mathcal{S}_H$. Let's now assume that $u_t(s_t) \geq V_{\pi,t}^{M_G}$ for $t = h+1, \cdots, H$. We prove by induction that it also holds for $t = h$,

$$
\begin{aligned}
u_h(s_h) &= \max_{a \in \mathcal{A}_h} \left\{ \sum_{v=1}^{n_h} r^{j_{h,v}}(x_h^v, y_h^v) + \sum_{s_{h+1} \in \mathcal{S}_{h+1}} \prod_{v=1}^{n_{h+1}} p^{i_{h,v}}(x_{h+1}^v | c_{h+1,a}^v) u_{h+1}(s_{h+1}) \right\} \\
&\geq \max_{a \in \mathcal{A}_h} \left\{ \sum_{v=1}^{n_h} r^{j_{h,v}}(x_h^v, y_h^v) + \sum_{s_{h+1} \in \mathcal{S}_{h+1}} \prod_{v=1}^{n_{h+1}} p^{i_{h,v}}(x_{h+1}^v | c_{h+1,a}^v) V_{\pi,h+1}^{M_G}(s_{h+1}) \right\} \\
&\geq \sum_{v=1}^{n_h} r^{j_{h,v}}(x_h^v, \pi(x_h^v | s_h)) + \sum_{s_{h+1} \in \mathcal{S}_{h+1}} \prod_{v=1}^{n_{h+1}} p^{i_{h,v}}(x_{h+1}^v | c_{h+1,\pi}^v) V_{\pi,h+1}^{M_G}(s_{h+1}) \\
&= \sum_{v=1}^{n_h} r^{j_{h,v}}(x_h^v, \pi(x_h^v | s_h)) + \mathbb{E}_\pi \left[ \sum_{t=h+1}^{H} \sum_{v=1}^{n_t} r^{j_{t,v}}(x_t^v, y_t^v) \right] \qquad \text{(by def. of } V^{M_G} \text{ in equation 3)} \\
&= V_{\pi,h}^{M_G}(s)
\end{aligned}
$$

where the notation $\mathbb{E}_\pi$ is used to explicitly mention that actions are selected according to the policy $\pi$.

**Proof of ii:** The second part is also shown by induction. We start by observing that the last step focuses only on maximising the immediate reward. So at time step $H$ it trivially holds that $u_H(s) = V_{*,H}^{M_G}(s)$ for all $s \in \mathcal{S}_H$. We then assume that $u_t(s) = V_{*,t}^{M_G}(s)$ for $t = h+1, \cdots, H$ and for all $s \in \mathcal{S}_t$. Then,

$$
\begin{aligned}
u_h(s) &= \max_{a \in \mathcal{A}_h} \left\{ \sum_{v=1}^{n_h} r^{j_{h,v}}(x_h^v, y_h^v) + \sum_{s' \in \mathcal{S}_{h+1}} \prod_{v=1}^{n_{h+1}} p^{i_{h,v}}(x_{h+1}^v | c_{h+1,a}^v) u_{h+1}(s') \right\} \\
&= \max_{a \in \mathcal{A}_h} \left\{ \sum_{v=1}^{n_h} r^{j_{h,v}}(x_h^v, y_h^v) + \sum_{s' \in \mathcal{S}_{h+1}} \prod_{v=1}^{n_{h+1}} p^{i_{h,v}}(x_{h+1}^v | c_{h+1,a}^v) V_{*,h+1}^{M_G}(s') \right\} \\
&= \sum_{v=1}^{n_h} r^{j_{h,v}}(x_h^v, \mu^*(x_h^v | s_h)) + \sum_{s' \in \mathcal{S}_{h+1}} \prod_{v=1}^{n_{h+1}} p^{i_{h,v}}(x_{h+1}^v | c_{h+1,*}^v) V_{*,h+1}^{M_G}(s') \\
&= V_{*,h}^{M_G}(s)
\end{aligned}
$$

Where the third line is obtained by the hypothesis that $u_t(s) = V_{*,t}^{M_G}(s)$ for $t \geq h+1$ and from the definition of $\mu^*$.

$\square$

## D   Proof of Theorem 5.1

We follow the proof structure of Osband et al. (2013), where the analysis focuses on a modified regret term that removes the dependency on $\mu^*$ and is equivalent to the original regret in expectation. For any episode $k$, we can write the modified regret as follows:

$$
\tilde{\Delta}_k = \sum_{s_1 \in \mathcal{S}_1} \rho(s_1)(V_{\mu_k,1}^{M_{G,k}}(s_1) - V_{\mu_k,1}^{M_G^*}(s_1)). \tag{18}
$$

where $M_G^*$ is the true DAMDP and $M_{G,k}$ is the DAMDP sampled at episode $k$.

The equivalence in expectation between the original regret $\Delta_k$ and the modified regret $\tilde{\Delta}_k$ is possible thanks to the following lemma, which is first presented in Osband et al. (2013, Lemma 1) and restated below in the DAMDP setting:

**Lemma D.1.** *If $f$ is the distribution of $M_G^*$ and $M_{G,k} \sim f$ then, for any $\sigma(D_k)-measurable$ function $g$,*

$$\mathbb{E}[g(M_G^*)|D_k] = \mathbb{E}[g(M_{G,k})|D_k]. \tag{19}$$

Where $\sigma(D_k)$ is the $\sigma$-algebra generated by all the data accumulated up to episode $k$, $D_k$.

This allows the following equivalence between the two different regret terms, $\mathbb{E}\left[\sum_{k=1}^K \Delta_k\right] = \mathbb{E}\left[\sum_{k=1}^K \tilde{\Delta}_k\right]$. Indeed, since $\Delta_k - \tilde{\Delta}_k = \sum_{s_1 \in \mathcal{S}_1} \rho(s_1)(V_{\mu_k,1}^{M_k}(s_1) - V_{\mu^*,1}^{M^*}(s_1))$, from Lemma D.1, we get that $\mathbb{E}[\Delta_k - \tilde{\Delta}_k|D_k] = 0$. Finally, by the tower rule $\mathbb{E}[\Delta_k - \tilde{\Delta}_k] = \mathbb{E}[\mathbb{E}[\Delta_k - \tilde{\Delta}_k|D_k]] = 0$.

To lighten the notation we now write $V_{\mu_k,i}^k$ for $V_{\mu_k,i}^{M_k}$. Working with the modified regret $\tilde{\Delta}$ (presented in equation 18) allows us to rewrite the modified regret in terms of Bellman error,

$$\mathbb{E}[\tilde{\Delta}_k|M_G^*, M_{G,k}] = \mathbb{E}\left[\sum_{i=1}^H (\mathcal{T}_{\mu,i}^{M_{G,k}} - \mathcal{T}_{\mu,i}^{M_G^*})V_{\mu,i+1}^k(s_{t_k+i})\bigg| M_G^*, M_{G,k}\right]. \tag{20}$$

To prove that equation 20 holds, we apply the Dynamic programming equation 7 inductively (note that we denote by $p_*^i(x|c)$ the atomic true transition distribution for nodes that belong to the equivalence class $[\tau_i]$):

$$
\begin{aligned}
(V_{\mu_k,1}^k - V_{\mu_k,1}^*)(s_{t_k+1}) &= (\mathcal{T}_{\mu_k,1}^k V_{\mu_k,2}^k - \mathcal{T}_{\mu_k,1}^* V_{\mu_k,2}^*)(s_{t_k+1}) \\
&= (\mathcal{T}_{\mu_k,1}^k - \mathcal{T}_{\mu_k,1}^*)V_{\mu_k,2}^k(s_{t_k+1}) + \sum_{s' \in \mathcal{S}_2} \prod_{v=1}^{n_2} p_*^{i_{2,v}}(x_2^v|c_2^v)(V_{\mu_k,2}^k - V_{\mu_k,2}^*)(s') \\
&= (\mathcal{T}_{\mu_k,1}^k - \mathcal{T}_{\mu_k,1}^*)V_{\mu_k,2}^k(s_{t_k+1}) + (V_{\mu_k,2}^k - V_{\mu_k,2}^*)(s_{t_k+2}) + d_{t_k+1} \\
&= \cdots \\
&= \sum_{h=1}^H (\mathcal{T}_{\mu_k,h}^k - \mathcal{T}_{\mu_k,h}^*)V_{\mu_k,h+1}^k(s_{t_k+h}) + \sum_{i=1}^\tau d_{t_k+h}
\end{aligned}
$$

for

$$d_{t_k+h} = \sum_{s' \in \mathcal{S}_{h+1}} \prod_{v=1}^{n_{h+1}} p_*^{i_{h,v}}(x_{h+1}^v|c_{i+1}^v)(V_{\mu_k,h+1}^k - V_{\mu_k,h+1}^*)(s') - (V_{\mu_k,h+1}^k - V_{\mu_k,h+1}^*)(s_{t_k+h+1}).$$

Since $\mathbb{E}[(V_{\mu_k,h+1}^k - V_{\mu_k,h+1}^*)(s_{t_k+h+1})|M_G^*, M_{G,k}] = \sum_{s' \in \mathcal{S}_{h+1}} \prod_{v=1}^{n_{h+1}}(p_*^{i_{h,v}}(x_{h+1}^v|c_{h+1,\mu_k}^v)(V_{\mu_k,h+1}^k - V_{\mu_k,h+1}^*)(s')$ the terms $d_{t_k+h}$ disappear in expectation (i.e. $\mathbb{E}[d_{t_k+h}|M_G^*, M_{G,k}] = 0$ for all $k$ and $h \in \{1, \cdots, H\}$) and so we obtain equation 20.

Since the regret has been rewritten as the sum of one-step Bellman errors (see equation 20), the next step is to show that as interactions with the environment are observed, the sampled DAMDPs, $M_{G,k}$, concentrate around the true DAMDP $M_G^*$. This is done in the following subsection.

### D.1 Confidence sets

For the remainder of this section, the notation is a bit more involved. The below list introduces or recalls the concepts and the notation used.

1. $G = (V, E)$ is an LDAG that encodes the underlying structure in the execution of a DAMDP episode. We denote the $i^{th}$ node of layer $t$ by $v_t^i$. We recall that each layer $t = 1, \cdots, H$ has $n_t$ nodes. The graph has a total of $|V| = \sum_{t=1}^{H} n_t = n$ nodes.

2. If two nodes $v_{t_1}^{i_1}$ and $v_{t_2}^{i_2}$, with $(i_1, t_1) \neq (i_2, t_2)$ have the same number of parents and their parents have the same atomic action space, they belong to the same transition equivalence class $[\tau_i]$. We denote by $d_i$ the number of parent (or incoming edges) of a node that belongs to equivalence class $[\tau_i]$. All nodes in $G$ belong to one of the $U_\tau$ equivalence classes, $[\tau_i]_{i=1}^{U_\tau}$. The number of nodes in $G$ that belong to the same equivalence class is denoted by $m_\tau^i$, and $\sum_{i=1}^{U_\tau} m_\tau^i = n$. For example, in Fig. 1, there are two different equivalence classes, $U_\tau = 2$, because there exist nodes with two parent nodes or a single parent node.

3. Each equivalence class $[\tau_i]$ with $i \in \{1, \cdots, U_\tau\}$, has a corresponding state-action space $\mathfrak{C}_i = \bigotimes_{k=1}^{d_i} (\mathcal{X} \times \mathcal{Y}_k)$ that contains all the possible values the transition context (of an element of the $i^{th}$ equivalence class $[\tau_i]$) can take. Recall that the atomic action space might depend on the node, so two nodes belong to the same equivalence class if they have the same number of parent nodes with the same atomic action spaces. This space consists of the atomic states and the atomic action values of each parent of the node. We can upper-bound the size of this context space by $(X\bar{Y})^{d_i}$, where $d_i$ is the number of parent nodes in the $i^{th}$ equivalence class, and $\bar{Y}$ is the size of the largest atomic action space.

4. Each equivalence class $[\tau_i]$ has a dedicated transition function $p^i(\cdot | c_i)$ for all $i \in \{1, \cdots, U_\tau\}$ and all possible transition context $c_i \in \mathfrak{C}_i$.

5. $U_r$ is the number of distinct atomic action sets. For example, in the case of the leaky maximum flow problem in Section 6.1 $U_r = 2$, the atomic action space is different if a node has a single out-going edge or two out-going edges. In the wind farm optimisation problem $U_r = 1$, regardless of the number of outgoing edges, the atomic action space of a node remains the same. The number of nodes in $G$ that have the atomic action set $j$ is denoted by $m_r^j$, for all $j \in \{1, \cdots, U_r\}$. Then, $\sum_{j=1}^{U_j} m_r^j = n$, where $n$ is the number of nodes in $G$.

6. For each distinct atomic action set $\{\mathcal{Y}\}_{j=1}^{U_r}$, we call the atomic reward context the associated atomic state-action space $\mathcal{Z}_j = \mathcal{X} \times \mathcal{Y}_j$. We denote by $z_j \in \mathcal{Z}_j$ an element of the reward context $j$.

7. For each atomic action set $\{\mathcal{Y}_j\}_{j=1}^{U_r}$ we define a corresponding atomic reward function $r^j(z_j)$.

8. Define $N_{t_k}^{\tau,i}(c)$ to count the number of times a node of the equivalence class $[\tau_i]$ has observed the transition context $c \in \mathfrak{C}_i$ during the first $t_k$ time steps, where $t_k$ indicates the time step at which episode k starts (i.e. $t_k = (k-1) * H + 1$). Sometimes, we are interested in monitoring the number of visits more closely. Then, $N_{t_k,v}^{\tau,i}(c)$ counts the number of time the transition context $c \in \mathfrak{C}_i$ given that the agent started episode $k$ and already observed the $v$ first nodes of $G$, with $v \in \{1, \cdots, n\}$. Nodes are indexed by layers, but the indexing within a layer is arbitrary and fixed before the learning starts. In Figure 1 leftmost plot, we show an example of an LDAG with a valid node indexing.

9. Define $N_{t_k}^{r,j}(z)$ to count the number of times a node has observed the reward context $z \in \mathcal{Z}_j$ during the first $t_k$ time steps for all $j \in [U_r]$ and for all $z \in \mathcal{Z}_j$. $N_{t_k,v}^{\tau,i}(z)$ counts the number of times the reward context $z$ was observed until the $v^{th}$ node of the execution of the $k^{th}$ episode, where the node indexing is the same as the one considered for $N_{t_k,v}^{\tau,i}(c)$ (see entry 8 of this list for more details).

The posterior sampling algorithm proceeds by sampling an atomic reward function for each possible action set $\{\mathcal{Y}_j\}_{j=1}^{U_r}$ and a transition function for each equivalence class $[\tau_i]$ for all $i \in \{1, \cdots, U_\tau\}$. The atomic transition distributions and the atomic reward distributions sampled at the beginning of episode $k$ (step 3. in Alg. 2) are denoted $\{p_k^i\}_{i=1}^{U_\tau}$ and $\{r_k^j\}_{j=1}^{U_r}$, respectively. The sampled DAMDP is then constructed by combining the atomic transition and reward function as dictated by the graph $G$. To show that the sampled DAMDPs, $M_{G,k}$, concentrate around the true DAMDP $M_G^*$, we construct confidence intervals around the

empirical atomic transition and reward functions. We define the confidence set of feasible DAMDPs that we may sample at episode $k$ as:

$$\mathcal{M}_G^k = \big\{ M : \|\hat{p}_k^i(\cdot|c) - p^i(\cdot|c)\|_1 \leq \beta_k^i(c) \; \forall c \in \mathfrak{C}_i, \; \forall i \in [U_\tau] \; \& $$
$$\|\hat{r}_k^j(z) - r^j(z)\| \leq \gamma_k^j(z) \; \forall z \in \mathcal{Z}_j \forall j \in [U_r] \big\}. \tag{21}$$

Where $\hat{p}_k^i$ denotes the empirical estimate at time step $k$ of the $i^{th}$ atomic transition function, $\hat{r}_k^j$ denotes the empirical estimate at time step $k$ of the $j^{th}$ reward function, $\beta_k^i(c)$ is the threshold for the error on the $i^{th}$ transition function, and $\gamma_k^j(z)$ is the error threshold for the error on the $j^{th}$ atomic reward function. We propose the following definition for the error thresholds:

$$\beta_k^i(c) = \sqrt{\frac{2}{N_{t_k}^{\tau,i}(c)} \left( X \log\left(\frac{2}{\delta}\right) + 2 d_i \log\left(\frac{X \bar{Y} m_t^i k}{\delta}\right) \right)} \quad \forall i \in \{1, \cdots, U_\tau\} \text{ and } \forall c \in \mathfrak{C}_i \tag{22}$$

$$\gamma_k^j(z) = \sqrt{\frac{\log\left(\frac{X \bar{Y} m_r^j k}{\delta}\right)}{\max(1, N_{t_k}^{r,j}(z))}} \quad \forall j \in \{1, \cdots, U_r\} \text{ and } \forall z \in \mathcal{Z}_j, \tag{23}$$

where $N_{t_k}^{r,j}(z)$ and $N_{t_k}^{\tau,i}(c)$ are defined above (elements 8 and 9 in the list).

### D.1.1 Analysis of the Confidence Sets

**Lemma D.2.** *For any $k \geq 1$, the true MDP $M_G^*$ belongs to the confidence set at episode $k$, $\mathcal{M}_G^k$, defined in equation 21 with probability:*
$$\mathbb{P}(M_G^* \notin \mathcal{M}_G^k) \leq \delta/k.$$

*Proof.* The $L_1$ deviation between the true atomic transition distribution $p^i(\cdot|c)$ and its empirical estimate $\hat{p}_k^i(\cdot|c)$ is bounded for any $\epsilon$ by (Weissman et al., 2003):

$$\mathbb{P}\left( \|\hat{p}_k^i(\cdot|c) - p^i(\cdot|c)\|_1 \geq \epsilon \right) \leq (2^X - 2) \exp\left( -\frac{n\epsilon^2}{2} \right) \quad \forall i \in \{1, \cdots, U_\tau\} \text{ and } \forall c \in \mathfrak{C}_i, \tag{24}$$

where $X$ is the number of distinct outcomes of $p(\cdot|c)$ and $n$ is a fixed number of samples.

We now define

$$\epsilon_\tau^i = \sqrt{\frac{2}{n} \log\left( \frac{2^X 2 U_\tau (X\bar{Y})^{d_i} m_t^i k^2}{\delta} \right)} \quad \text{where} \quad \epsilon_\tau^i \leq \sqrt{\frac{2}{n} \left( X \log\left(\frac{2}{\delta}\right) + 2 d_i \log\left(\frac{2 U_\tau X \bar{Y} m_t^i k}{\delta}\right) \right)}.$$

This upper bound on $\epsilon_\tau^i$ gives the exact expression of $\beta_k^i(c)$ defined in equation 22.

We can, therefore, show that the confidence bound in equation 21 holds with high probability. In particular, for the atomic transition probability of a given context $c \in \mathfrak{C}_i$ and fixed number of visits $n$ we have:

$$\mathbb{P}\left( \|\hat{p}_k^i(\cdot|c) - p^i(\cdot|c)\|_1 \geq \beta_k^i(c) \right) = \mathbb{P}\left( \|\hat{p}_k^i(\cdot|c) - p^i(\cdot|c)\|_1 \geq \sqrt{\frac{2}{n} \left( X \log\left(\frac{2}{\delta}\right) + 2 d_i \log\left(\frac{2 U_\tau X \bar{Y} m_t^i k}{\delta}\right) \right)} \right)$$

$$\leq 2^X \exp\left( -\frac{n}{2} (\epsilon_\tau^i)^2 \right)$$

$$= 2^X \exp\left( -\frac{n}{2} \frac{2}{n} \log\left( \frac{2^X 2 U_\tau (X\bar{Y})^{d_i} m_t^i k^2}{\delta} \right) \right)$$

$$= \frac{\delta}{2 U_\tau (X\bar{Y})^{d_i} m_t^i k^2}.$$

Now, with a fixed number of samples $n$, the deviation between the true atomic expected reward and the empirical atomic mean reward is bounded by Hoeffding's inequality for any $\epsilon_r^j > 0$:

$$\mathbb{P}\left(|\hat{r}_k^j(z) - r^j(z)| \geq \epsilon_r^j\right) \leq 2\exp(-2n(\epsilon_r^j)^2) \quad \forall j \in \{1, \cdots, U_r\} \text{ and } \forall z \in \mathcal{Z}_j. \tag{25}$$

We define

$$\epsilon_r^j = \sqrt{\frac{1}{2n}\log\left(\frac{4U_r X\bar{Y}m_r^j k^2}{\delta}\right)} \quad \text{and note that} \quad \epsilon_r^j \leq \sqrt{\frac{1}{n}\log\left(\frac{4U_r X\bar{Y}m_r^j k}{\delta}\right)}.$$

Again, this upper bound on $\epsilon_r^j$ will be used to determine the exact expression of $\gamma_k^j(z)$.

Combining Hoeffdings inequality with our definition of $\epsilon_r^j$, we get:

$$\mathbb{P}\left(|\hat{r}_k^j(z) - r^j(z)| \geq \gamma_k^j(z)\right) = \mathbb{P}\left(|\hat{r}_k^j(z) - r^j(z)| \geq \sqrt{\frac{1}{n}\log\left(\frac{4U_r X\bar{Y}m_r^j k}{\delta}\right)}\right) \tag{26}$$

$$\leq 2\exp(-2n(\epsilon_r^j)^2) \tag{27}$$

$$= 2\exp\left(-\frac{2}{n}\frac{n}{2}\log\left(\frac{4U_r X\bar{Y}m_r^j k^2}{\delta}\right)\right) \tag{28}$$

$$= \frac{\delta}{2U_r X\bar{Y}m_r^j k^2} \tag{29}$$

To compute the probability of interest $\mathbb{P}(M_G^* \notin \mathcal{M}_G^k)$, we need to compute the union bound by summing over all possible numbers of visits. It is important to note that in the same layer, more than one node might observe a reward for the same *atomic state-action* pair $z = (x, y)$. It is also possible that in the same layer, more than one node observes the same transition context $c$. As a consequence, the number of observations can increase by more than a unit per time step. If we perform $k$ episodes, the number of visits for a specific reward context $j$ might range between 0 and $km_r^j$, where $m_r^j$ is the number of occurrences of reward architecture $j$ in $G$ (see element 5 of the above list for a more complete definition). Similarly, there are $m_\tau^i$ nodes that belong to the equivalence class $[\tau_i]$, so a given transition context $c_i$ can be observed up to $km_\tau^i$ times (see element 2 of the above list for a more complete definition). Now, computing the union bound by summing over all possible numbers of visits, we obtain the following:

$$\mathbb{P}\left(|\hat{r}_k^j(z) - r^j(z)| \geq \sqrt{\frac{\log\left(\frac{4U_r X\bar{Y}m_r^j k}{\delta}\right)}{\max(1, N_{t_k}^{r,j}(z_j))}}\right) \leq \sum_{n=1}^{m_r^j k}\frac{\delta}{2U_r X\bar{Y}m_r^j k^2} < \frac{\delta}{2U_r X\bar{Y}k} \tag{30}$$

$$\mathbb{P}\left(\|\hat{p}_k^i(\cdot|c) - p^i(\cdot|c)\|_1 \geq \sqrt{\frac{2}{\max(1, N_{t_k}^{\tau,i}(c_i))}\left(X\log\left(\frac{2}{\delta}\right) + 2d_i\log\left(\frac{2U_\tau X\bar{Y}m_t^i k}{\delta}\right)\right)}\right)$$

$$\leq \sum_{n=1}^{m_t^i k}\frac{\delta}{2U_\tau(XY)^{d_i}m_t^i k^2} < \frac{\delta}{2U_\tau(X\bar{Y})^{d_i}k}$$

This allows us to define $\beta_k^i(c)$ (introduced in equation 21) for all $i \in \{1, \cdots, U_\tau\}$ and for all $c \in \mathfrak{C}_i$:

$$\beta_k^i(c) := \sqrt{\frac{2}{\max(1, N_{t_k}^{\tau,i}(c))}\left(X\log\left(\frac{2}{\delta}\right) + 2d_i\log\left(\frac{2U_\tau X\bar{Y}m_t^i k}{\delta}\right)\right)} \tag{31}$$

and $\gamma_k^j(z)$ for all $j \in \{1, \cdots, U_r\}$ and for all $z \in \mathcal{Z}_i$, as

$$\gamma_k^j(z) := \sqrt{\frac{\log\left(\frac{4U_r X\bar{Y} m_r^j k}{\delta}\right)}{\max(1, N_{t_k}^{r,j}(z))}}. \tag{32}$$

Summing the probabilities over all possible reward contexts $z \in \mathcal{Z}_j$ (the number of which is bounded by $X\bar{Y}$) and transition context $c \in \mathfrak{C}_i$ (the number of which is bounded by $(X\bar{Y})^{d_i}$) we get

$$\mathbb{P}(M_G^* \notin \mathcal{M}_G^k) = \sum_{j=1}^{U_r} \sum_{z \in \mathcal{Z}_j} \mathbb{P}(|\hat{r}_k^j(z) - r^j(z)| < \gamma_k^j(z)) + \sum_{i=1}^{U_\tau} \sum_{c \in \mathfrak{C}_i} \mathbb{P}(\|\hat{p}_k^i(\cdot|c) - p^i(\cdot|c)\| < \beta_k^i(c)) \tag{33}$$

$$\leq \sum_{j=1}^{U_r} \sum_{z \in \mathcal{Z}_j} \frac{\delta}{2U_r X\bar{Y} k} + \sum_{i=1}^{U_\tau} \sum_{c \in \mathcal{C}_i} \frac{\delta}{2U_\tau (X\bar{Y})^{d_i} k} \leq \frac{\delta}{k} \tag{34}$$

as desired. $\qquad\square$

## D.2 Upper Bounding the Regret

The regret can be upper bounded by the sum of the errors made at each node.

**Lemma D.3.** *If $M_{G,k}$ and $M_G^*$ belong to the confidence set at time step $k$, $\mathcal{M}_G^k$, then the one-step Bellman error is upper bounded by the size of the confidence interval:*

$$|(\mathcal{T}_{\mu_k,h}^k - \mathcal{T}_{\mu_k,h}^*)V_{\mu_k,h+1}^k(s_{h+1})| \leq H \min\{\sum_{v=1}^{n} \beta_k^{i_v}(c_k^v) + \gamma_k^{j_v}(z_k^v), 1\} \tag{35}$$

*Proof.* Error in the atomic Bellman operator has the most impact if the value function is high; we can then directly upper bound the value function with $H$.

$$|(\mathcal{T}_{\mu_k,h}^k - \mathcal{T}_{\mu_k,h}^*)V_{\mu_k,h+1}^k(s_{h+1})| \leq H|(\mathcal{T}_{\mu_k,h}^k - \mathcal{T}_{\mu_k,h}^*)| \tag{36}$$

$$\leq H\left(\left|\sum_{v=1}^{n_h} r_k^{j_v}(x_k^v|c_{h,\mu}^v) + \prod_{v=1}^{n_{h+1}} p_k^{i_v}(x_{h+1}^v|c_{h+1,\mu}^v) - \sum_{v=1}^{n_h} r_*^{j_v}(x_k^v|c_{h,\mu}^v) + \prod_{v=1}^{n_{h+1}} p_*^{i_v}(x_{h+1}^v|c_{h+1,\mu}^v)\right|\right) \tag{37}$$

$$\leq H\left(\left|\sum_{v=1}^{n_h} r_k^{j_v}(x_k^v|c_{h,\mu}^v) - \sum_{v=1}^{n_h} r_*^{j_v}(x_k^v|c_{h,\mu}^v)\right| + \left|\prod_{v=1}^{n_{h+1}} p_k^{i_v}(x_{h+1}^v|c_{h+1,\mu}^v) - \prod_{v=1}^{n_{h+1}} p_*^{i_v}(x_{h+1}^v|c_{h+1,\mu}^v)\right|\right) \tag{38}$$

$$\leq H\left(\left|\sum_{v=1}^{n_h} r_k^{j_v}(x_k^v|c_{h,\mu}^v) - \sum_{v=1}^{n_h} r_*^{j_v}(x_k^v|c_{h,\mu}^v)\right| + \sum_{v=1}^{n_{h+1}} \left|p_k^{i_v}(x_{h+1}^v|c_{h+1,\mu}^v) - p_*^{i_v}(x_{h+1}^v|c_{h+1,\mu}^v)\right|\right) \tag{39}$$

$$\leq H \min\left\{\sum_{v=1}^{n_h} \gamma_k^{j_v}(z_{h,\mu}^v) + \beta_k^{i_v}(c_{h+1,\mu}^v), 1\right\} \tag{40}$$

Equation 37 is directly obtained using the definition of the atomic Bellman operator equation 7. Then equation 38 is obtained by grouping together the terms depending on the atomic reward functions and the one depending on the atomic transition functions. To obtain equation 39 we use the following inequality:

$$\prod_{i=1}^{n} a_i - \prod_{i=1}^{n} b_i \leq \sum_{i=1}^{n} |a_i - b_i|, \quad \forall a_i, b_i \leq 1.$$

Finally, in equation 40, we upper-bound the error in the transition and reward function at each node with the width of the corresponding confidence interval. Note that since the reward at each time step is bounded in $[0, 1]$, the sum of all the nodes' errors in a given layer is upper-bounded by 1. $\qquad\square$

With this upper bound on the Bellman error, we are now equipped to upper-bound the regret; we first decompose it as follows, observing that $\tilde{\Delta}_k \leq H$ for any $k \in \{1, \cdots, K\}$. Hence,

$$\sum_{k=1}^{K} \tilde{\Delta}_k \leq \sum_{k=1}^{K} \tilde{\Delta}_k \mathbb{1}_{\{M_G^k, M_G^* \in \mathcal{M}_G^k\}} + H \sum_{k=1}^{K} \left[ \mathbb{1}_{\{M_G^* \notin \mathcal{M}_G^k\}} + \mathbb{1}_{\{M_G^k \notin \mathcal{M}_G^k\}} \right].$$

By Lemma D.1, $\mathbb{E}[\mathbb{1}_{\{M_{G,k} \notin \mathcal{M}_G^k\}} | D_{t_k}] = \mathbb{E}[\mathbb{1}_{\{M_G^* \notin \mathcal{M}_G^k\}} | D_{t_k}]$. Additionally, setting $\delta = \frac{1}{K}$ in Lemma D.2 shows the true DAMDP $M_G^*$ does not belong to the confidence set at time step $k$, $\mathcal{M}_G^k$, with probability $\mathbb{P}(M_G^* \notin \mathcal{M}_G^k) < \frac{1}{K}$. Then,

$$\mathbb{E}\left[ \sum_{k=1}^{K} \tilde{\Delta}_k \right] \leq \mathbb{E}\left[ \sum_{k=1}^{K} \tilde{\Delta}_k \mathbb{1}_{\{M_{G,k}, M_G^* \in \mathcal{M}_G^k\}} \right] + 2H \sum_{k=1}^{K} \mathbb{P}(M^* \notin \mathcal{M}_k) \tag{41}$$

$$\leq \mathbb{E}\left[ \sum_{k=1}^{K} \mathbb{E}[\tilde{\Delta}_k | M_G^*, M_{G,k}] \mathbb{1}_{\{M_{G,k}, M_G^* \in \mathcal{M}_G^k\}} \right] + 2H \tag{42}$$

$$\leq \mathbb{E}\left[ \sum_{k=1}^{K} \sum_{h=1}^{H} |(\mathcal{T}_{\mu_k,h}^k - \mathcal{T}_{\mu_k,h}^*) V_{\mu_k,h+1}^k(s_{t_k+1})| \mathbb{1}_{\{M_{G,k}, M_G^* \in \mathcal{M}_G^k\}} \right] + 2H \tag{43}$$

$$\leq H \mathbb{E}\left[ \sum_{k=1}^{K} \sum_{h=1}^{H} \min\left\{ \sum_{v=1}^{n_h} \beta_k^{i_v}(c_k^v) + \gamma_k^{j_v}(z_k^v), 1 \right\} \right] + 2H, \tag{44}$$

where Eq. 42 is obtained by applying the tower property and noting that $\mathcal{M}_G^k$ is measurable if $M_{G,k}$ is known. Equation 43 is the definition of the modified regret term (see equation 18). Finally, equation 44 is a direct consequence of Lemma D.3. We also denote by $n$ the number of nodes in $G$; we use $i_v$ to denote the equivalence class of the node $v$ and $j_v$ to denote the atomic action set available at node $v$. The transition context observed during episode $k$ at node $v$ is denoted by $c_k^v \in \mathfrak{C}_{i_v}$, and the reward context observed at node $v$ during the $k^{th}$ episode is denoted by $z_k^v \in \mathcal{Z}_{j_v}$.

The contribution to the regret incurred by errors in the reward function $r_k^j$ can be upper-bounded by the sum of $\gamma_k^j$ for every episode and every node $v \in V_j$, where $V_j$ is the set of nodes that exhibit the reward architecture $j$. Then, for a single reward architecture $j$ we have:

$$\sum_{k=1}^{K} \sum_{v \in V_j} \gamma_k^j(z_k^v) = \sum_{k=1}^{K} \sum_{v \in V_j} \gamma_k^j(z_k^v) \mathbb{1}_{\{N_{t_k}^{r,j}(z_k^v) \leq m_r^j\}} + \sum_{k=1}^{K} \sum_{v \in V_j} \gamma_k^j(z_k^v) \mathbb{1}_{\{N_{t_k}^{r,j}(z_k^v) > m_r^j\}}$$

$$\leq \sum_{k=1}^{K} \sum_{v \in V_j} \mathbb{1}_{\{N_{t_k}^{r,j}(z_k^v) \leq m_r^j\}} + \sum_{k=1}^{K} \sum_{v \in V_j} \mathbb{1}_{\{N_{t_k}^{r,j}(z_k^v) > m_r^j\}} \sqrt{\frac{\log\left(\frac{4U_r X \bar{Y} m_r^j k}{\delta}\right)}{\max(1, N_{t_k}^{r,j}(z_k^v))}}.$$

Where $z_k^v \in \mathcal{Z}_j$ is used to denote the reward context observed in the $v^{th}$ node of $G$ during the $k^{th}$ episode.

Consider a fixed $z_k^v = z$. The case where $N_{t_k}^{r,j}(z) \leq m_r^j$ happens less than $2m_r^j$ times for each $z \in \mathcal{Z}_j$ can be upper bounded as follows $\sum_{k=1}^{K} \sum_{v \in V_j} \mathbb{1}_{\{N_{t_k}^{r,j}(z) \leq m_r^j\}} \leq 2m_r^j X \bar{Y}$. Now, let's suppose that $N_{t_k}^{r,j}(z) > m_r^j$ for a $z \in \mathcal{Z}_j$. In the $k^{th}$ episode, for any node $v \in V_j$, we have $N_{t_k,n}^{r,j}(z) + 1 \leq N_{t_k}^{r,j}(z) + m_r^j \leq 2N_{t_k}^{r,j}(z)$. Note that the number of occurrences of a specific reward architecture in each layer depends on $G$, but we know that it occurs $m_r^j$ times over an episode.

We can then bound the following ratio:

$$\sum_{k=1}^{K}\sum_{v\in V_j}\sqrt{\frac{\mathbb{1}_{\{N_{t_k}^{r,j}(z_{k,v})>m_r^j\}}}{N_{t_k}^{r,j}(z_{k,v})}} \leq \sum_{k=1}^{K}\sum_{v\in V_j}\sqrt{\frac{2}{N_{t_k,v}^{r,j}(z_{k,v})+1}} \tag{45}$$

$$= \sqrt{2}\sum_{t=1}^{Km_j^r}(N_t^{r,j}(z_t)+1)^{-1/2} \tag{46}$$

$$\leq \sqrt{2}\sum_{z\in\mathcal{Z}_j}\sum_{b=1}^{N_{T_j}^{r,j}(z)}b^{-1/2} \quad (\text{with } T_j = Km_r^j+1) \tag{47}$$

$$\leq \sqrt{2}\sum_{z\in\mathcal{Z}_j}\int_0^{N_{T_j}^{r,j}(z)}x^{-1/2}dx \tag{48}$$

$$\leq \sqrt{2X\bar{Y}\sum_{z\in\mathcal{Z}_j}N_{T_j}^{r,j}(z)} = \sqrt{2X\bar{Y}Km_r^j}. \tag{49}$$

Equation 46 rewrites the two sums as a single one, where now, each index $t$ in $N_t^{r,j}$ uniquely encodes a pair $(t_k, n)$. We obtain equation 47 by considering all possible reward contexts $z \in \mathcal{Z}_j$ and the total number of times this specific reward context was visited. In equation 48 we use the fact that the sum $\sum_{i=n+1}^{N}x^{-1/2}$ is upper bounded by $\int_n^N x^{-1/2}dx$. Lastly, equation 49 is obtained by Cauchy-Schwartz inequality.

With the same approach, we can bound the regret incurred by the error in the $i^{th}$ transition function,

$$\sum_{k=1}^{K}\sum_{v\in V_i}\beta_k^i(c_k^v) \leq \sum_{k=1}^{K}\sum_{v\in V_i}\mathbb{1}_{\{N_{t_k}^{\tau,i}(c_k^v)\leq m_\tau^i\}} + \sum_{k=1}^{K}\sum_{v\in V_i}\mathbb{1}_{\{N_{t_k}^{\tau,i}(c_k^v)>m_\tau^i\}}\beta_k^i(c_k^v),$$

where $V_i$ denotes all the nodes of the equivalence class $[\tau_i]$. For any $c \in \mathfrak{C}_i$, the case where $N_{t_k}^{\tau,i}(c) \leq m_\tau^i$ happens less than $2m_\tau^i$ times. Given that $|\mathfrak{C}_i| \leq (X\bar{Y})^{d_i}$ we can upper-bound $\sum_{k=1}^{K}\sum_{v\in V_i}\mathbb{1}_{\{N_{t_k}^{\tau,i}(c)\leq m_\tau^i\}} \leq 2m_\tau^i(X\bar{Y})^{d_i}$. For a given transition context $c$ and episode $k$ when $N_{t_k}^{\tau,i} > m_\tau^i$, we have that for any node $V \in V_i$, $N_{t_k,v}^{\tau,i}(c)+1 \leq N_{t_k}^{\tau,i}(c)+m_\tau^i \leq 2N_{t_k}^{\tau,i}(c)$.

We can then bound the following ratio:

$$\sum_{k=1}^{K}\sum_{v\in V_i}\sqrt{\frac{\mathbb{1}_{\{N_{t_k}^{\tau,i}(c_k^v)>m_\tau^i\}}}{N_{t_k}^{\tau,i}(c_v^v)}} \leq \sum_{k=1}^{K}\sum_{v\in V_i}\sqrt{\frac{2}{N_{t_k,v}^{\tau,i}(c_k^v)+1}} \tag{50}$$

$$= \sqrt{2}\sum_{t=1}^{Km_i^\tau}(N_t^{\tau,i}(c_t)+1)^{-1/2} \tag{51}$$

$$\leq \sqrt{2}\sum_{c\in\mathfrak{C}_i}\sum_{b=1}^{N_{T_i}^{\tau,i}(c)}b^{-1/2} \quad (\text{with } T_i = Km_\tau^i+1) \tag{52}$$

$$\leq \sqrt{2}\sum_{c\in\mathfrak{C}_i}\int_0^{N_{T_i}^{\tau,i}(c)}x^{-1/2}dx \tag{53}$$

$$\leq \sqrt{2(X\bar{Y})^{d_i}\sum_{c\in\mathfrak{C}_i}N_{T_i}^{\tau,i}(c)} = \sqrt{2(X\bar{Y})^{d_i}Km_\tau^i}. \tag{54}$$

Where all steps here are obtained following the same reasoning as in equation 45 toequation 49.

Because the rewards at every time step (or layer) are bounded in $[0, 1]$, the total regret consists of the following:

$$\min\left\{H\sum_{k=1}^{K}\sum_{h=1}^{H}\min\left\{\sum_{v=1}^{n_h}\gamma_k^{i_v}(z_k^v)+\beta_k^{i_v}(c_k^v),1\right\},T\right\} \tag{55}$$

$$\leq\min\left\{H\sum_{j=1}^{U_r}2m_r^j X\bar{Y}+\sqrt{2X\bar{Y}Km_r^j\log(4U_r X\bar{Y}m_r^j K)}\right.$$

$$\left.+H\sum_{i=1}^{U_\tau}2m_\tau^i(X\bar{Y})^{d_i}+\sqrt{2(X\bar{Y})^{d_i}Km_\tau^i\big(X\log(\frac{2}{\delta}+2d_i\log(2U_\tau X\bar{Y}m_\tau^i)))},KH\right\} \tag{56}$$

$$\leq\min\left\{H\sum_{j=1}^{U_r}2m_r^j X\bar{Y}+\sqrt{2X\bar{Y}Km_r^j\log(4U_r X\bar{Y}m_r^j K)},KH\right\}$$

$$+\min\left\{H\sum_{i=1}^{U_\tau}2m_\tau^i(X\bar{Y})^{d_i}+\sqrt{2(X\bar{Y})^{d_i}Km_\tau^i\big(X\log(\frac{2}{\delta}+2d_i\log(2U_\tau X\bar{Y}m_\tau^i)))},KH\right\} \tag{57}$$

To complete the proof, we note that $\min(a+b,c)\leq\sqrt{ac}+b$ holds for $a,b,c>0$. We then apply this inequality twice in equation 57.

$$\text{Equation } 57 \leq \sqrt{KH\sum_{j=1}^{U_r}2m_r^j X\bar{Y}H}+\sum_{j=1}^{U_r}\sqrt{2X\bar{Y}Km_r^j\log(4U_r X\bar{Y}m_r^j K)H}$$

$$+\sqrt{KH\sum_{i=1}^{U_\tau}2m_\tau^i(X\bar{Y})^{d_i}H}+\sum_{i=1}^{U_\tau}\sqrt{2(X\bar{Y})^{d_i}HKm_\tau^i(X+2d_i\log(2U_\tau X\bar{Y}m_\tau^i K))}$$

$$\leq H\sum_{j=1}^{U_r}\sqrt{2X\bar{Y}Km_r^j\log(4U_r X\bar{Y}m_r^j K)}+\sum_{i=1}^{U_\tau}H\sqrt{2(X\bar{Y})^{d_i}Km_\tau^i(X+2d_i\log(2U_\tau X\bar{Y}m_\tau^i K))}$$

This gives us the result obtained in Eq. 10.

# E  Regret of PSRL on the *full* MDP

A natural baseline to our approach is to consider the Posterior Sampling for Reinforcement Learning (Osband et al., 2013) algorithm. This algorithm can solve any DAMDP by considering the MDP built from the DAMDP's atomic components. In that case, the algorithm will directly learn the *full state* transition distribution $P$ and the reward function $R$.

To simplify the analysis, we upper bound the state space at each time step $t$, $\mathcal{S}_t\subseteq\mathcal{X}^{N_{max}}$, with the largest state space in the MDP, note that $N_{max}$ denotes the number of nodes in the largest layer. Similarly, we upper bound the *full action* space at each time step $t$, $\mathcal{A}\subseteq\bar{\mathcal{Y}}^{N_{max}}$.

If we apply the results of Osband et al. (2013) to the proposed MDP, we get a regret of

$$O\left(H\mathcal{X}^{N_{max}}\sqrt{\bar{\mathcal{Y}}^{N_{max}}T\log((\mathcal{X}\bar{\mathcal{Y}})^{N_{max}}T)}\right) \tag{58}$$

# F  Computational complexity of PSGRL

This section provides an analysis of the computational complexity of running PSGRL. In PSGRL (see Alg. 2), there are two sub-routines with non-trivial runtime, i.e. "Sample $M_{G,k}\sim f(\cdot|D_{t_k})$" and "Compute $\mu^{M_{G,k}}$ using Algorithm 1".

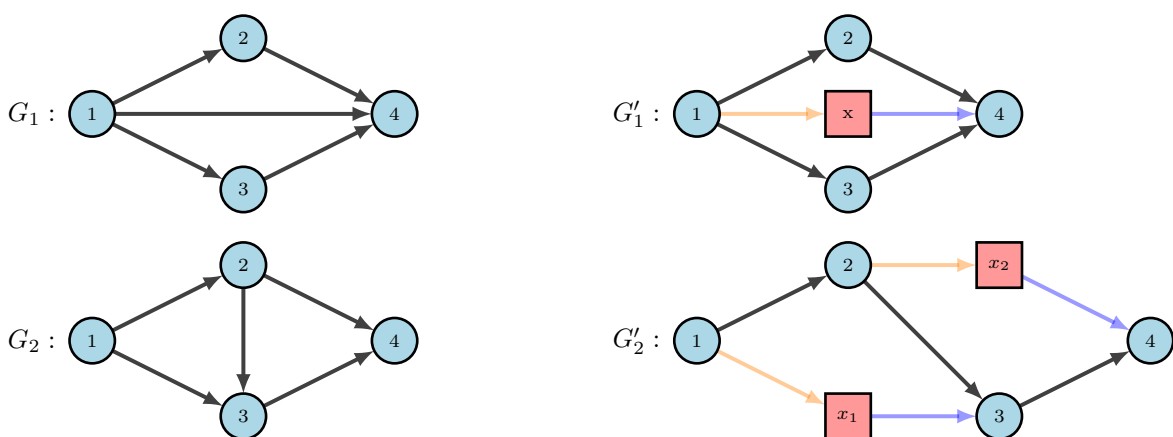

Figure 5: Illustration of how to transform a directed acyclic graph (on the left) into a layered directed acyclic graph (on the right)

First, let's consider the operations when the algorithm "samples $M_{G,k} \sim f(\cdot|D_{t_k})$". At this point, the algorithm needs to sample the *atomic* transition and reward functions, $\{p^i\}_{i=1}^{U_\tau}$ and $\{r^j\}_{j=1}^{U_r}$. In the case of the transition function, the algorithm requires sampling the parameters for every equivalence class $i \in [U_\tau]$, every context $c \in \mathfrak{C}_i$ and for any potential next atomic state $x \in \mathcal{X}$. This consists of $\mathcal{O}(U_\tau X^{2d_{\max}} \bar{Y}^{d_{\max}})$ operations. Similar reasoning can be used regarding the sampling of the reward functions. Resulting in a combined number of operation to be $\mathcal{O}(U_\tau X^{2d_{\max}} \bar{Y}^{d_{\max}} + U_r X \bar{Y})$.

Second, the planning phase does not benefit from any speed up and is equivalent to the computational complexity of running backward induction on the considered non-stationary MDP, which consists of $\mathcal{O}(\sum_{t=1}^{H} S_t A_t)$ operations.

In conclusion the computational complexity of running PSGRL for every episode $k \in [K]$ consists of $\mathcal{O}(\sum_{t=1}^{H} S_t A_t + U_\tau X^{2d_{\max}} \bar{Y}^{d_{\max}} + U_r X \bar{Y})$ operations.

## G  Additional properties of DAMDP

*Remark* G.1. Any DAMDP with an arbitrary DAG, $G$, has an equivalent DAMDP with a layered directed acyclic graph, $G'$.

We now show how a LDAG, $G'$, can be constructed from a DAG, $G$. In particular, we focus on two possible transformations that are illustrated in Figure 5, that is, when an edge spans more than one layer (first row of Fig. 5) or when we have a transversal edge, i.e. an edge that connects two nodes that belong to the same layer, (second row of Fig. 5).

Note that both illustrations Figure 5 suggest that to transform a DAG into LDAG, we need to add artificial nodes (represented by a red square) and artificial edges (coloured in blue and orange). The artificial node does not directly increase the complexity of the problem, as both its atomic state and atomic action will be identical to the atomic state and atomic action of its unique parent node. The additional edges do not increase the problem complexity as when we remove an "illegal" edge with a single orange edge and a single blue edge. In particular, it leaves the number of equivalence classes $U_\tau$ unchanged since all incoming illegal edges were replaced by a single blue edge, and it leaves the number of action spaces unchanged as well as all outgoing illegal edges are replaced by an orange edge. Since those transformations are always possible and leave the DAMDP characteristics unchanged, they have no impact on the algorithm complexity measure presented in Theorem 5.1.

While this argument is reasonable in problems of the type of "the leaky maximum flow" problem, this will not necessarily hold for more complex problems such as the wind farm optimization problem. Indeed, the LDAG assumption allowed us to assume all the layers were spaced uniformly in the field. Hence, the

transition context did not necessarily have to include information about the relative distance between two wind turbines, for example. If we would like to consider such scenarios, then the transformations are not trivial and will have repercussions on the algorithm's regret.

## H   Additional experiments

In Figure 6, we present an additional set of leaky maximum flow experiments. We consider an alternative graph structure presented in the first row of Figure 6. Observing the respective performance of PSRL and PSGRL, we can see that as the graph complexity increases, the benefit of PSGRL increases as well. These results are similar to the experiment presented in the main paper; we just revisit this experiment considering a different network architecture. In the leftmost case, we see no benefit in using PSGRL as we have a single node per layer. However, as we increase the central grid of nodes, creating multiple nodes that belong to the same equivalence class, the benefit of PSGRL becomes more evident.

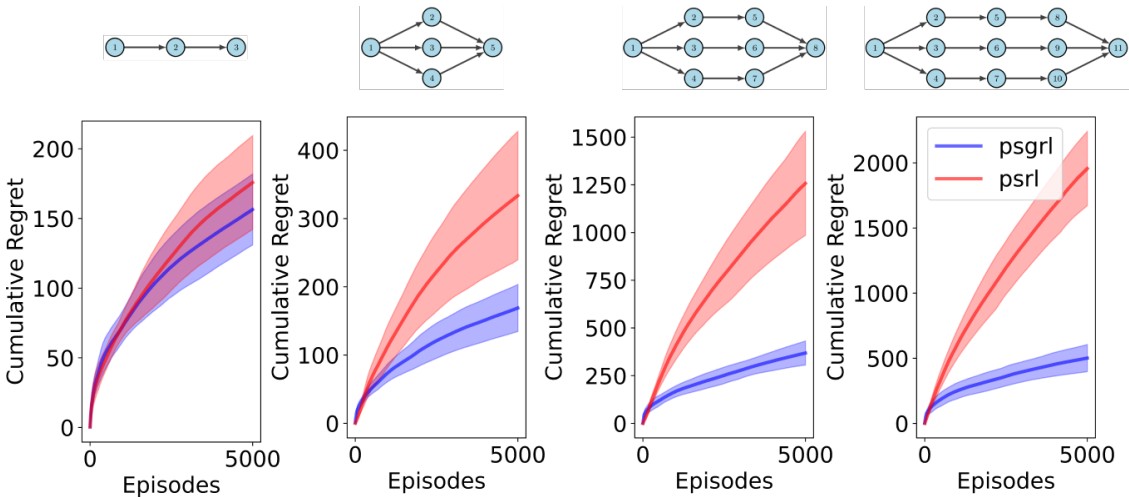

Figure 6: The first row depicts the graph that governs the DAMDP. The second row shows the learning curve for both algorithms considered. PSRL, which ignores the latent graphical structure and PSGRL, which leverages the graphical structure. The left-most plot shows the performance obtained on a simple chain graph. As expected, the performance for both algorithms is similar. Looking at the remaining plots, it becomes clear that as we increase the complexity of the graph, the benefit of PSGRL becomes self-evident.

