# OpenReview forum: "Posterior Sampling for Reinforcement Learning on Graphs"
_TMLR — Accepted by TMLR_

### Review · Reviewer_6dxW · 2024-11-16

**Summary Of Contributions:**

The authors investigate a specific structural assumption of MDPs based on DAGs. They show that it is possible to apply dynamic programming and posterior sampling in this setting where the main advantage is that the resulting algorithm can learn more efficiently by exploiting the known graphical structure. A rigorous theoretical analysis of the approach is included.

**Audience:**

Yes

**Broader Impact Concerns:**

No concerns.

**Claims And Evidence:**

Yes

**Requested Changes:**

# Changes
- Could you run the experiments of PSRL again with the graph structure encoded in the prior distributions? In other words, if there is a known edge between `s` and `s'` you can add pseudo counts in the corresponding distributions.
-Could you add a brief section regarding the practical computational cost of running PSGRL?

**Strengths And Weaknesses:**

# Strengths
- Utilizing prior information to improve reinforcement learning algorithm represents a key challenge. The proposed method is a good step into this direction and is also supported with a real-world example.
- The presentation is very clear and the paper is well structured.
- The theoretical proof is well presented and necessary details are well explained.

# Weaknesses
- In the experiment, the authors compare PSGRL that has access to the graph structure against PSRL that has no prior information. This comparison feels a bit unfair considering that it is possible to partially encode the graph structure in PSRL via the Dirichlet prior distributions.
- The MDPS used for the experiments seem to all have a small state space. Is there any reason for this? How is the scalability of the algorithm in practice? What are the computation challenges?

---

> ### Author Response · Authors · 2024-12-17
> **Reply**
>
> Many thanks for your feedback and enthusiasm about our paper. We are grateful that you recognise that we are working on a “key challenge” in RL and that our delivery is “very clear and well-structured”.
>
> We now comment on the requested changes and weaknesses.
>
> i.  The prior of PSRL does not completely ignore the structure. The algorithm has to solve a non-stationary MDP with a varying state space; the prior we use knows the state dimensionality at each time step. Therefore, it already incorporates some information about the structure. While it would be interesting to consider incorporating more structural information into the prior, it is not clear how this could be done in the proposed setup. PSRL operates on the “full” MDP; with this level of resolution, there are no connections between the full states $s$ and $s’$, so it is not possible to encode this information into a prior. The graphical relationship can only be modelled when we consider the atomic representation of states as in our PSGRL algorithm. Our experiments confirm our theoretical findings that when there is an underlying graphical structure, our algorithm can exploit this to benefit from improved learning efficiency.
>
> ii. We agree that we are missing a discussion about the computational complexity of our algorithm. We will add a description of our algorithm's computational complexity; thanks for the suggestion!
>
> We hope the above clarifications will reinforce your confidence that our paper is valuable to the community.
>
> Sincerely,
>
> The authors

---

### Review · Reviewer_yEM6 · 2024-12-06

**Summary Of Contributions:**

The paper introduces Directed Acyclic Markov Decision Processes (DAMDPs) as a subclass of Markov Decision Processes (MDPs) that leverage directed acyclic graph (DAG) structures in state and action spaces.

They introduce

  - A dynamic programming algorithm is developed to find optimal policies in known DAMDPsand a posterior sampling-based reinforcement learning algorithm is proposed for DAMDPs.

 - An upper bound on Bayesian regret for PSGRL, demonstrating improved efficiency over standard reinforcement learning approaches that ignore graph structures.

Experimental results on graph flow optimization and wind farm yield optimization showcase significant performance improvements of PSGRL over baseline methods.

**Audience:**

Yes

**Claims And Evidence:**

Yes

**Requested Changes:**

1.   Include comparisons with additional reinforcement learning algorithms beyond PSRL, particularly model-based approaches or methods handling structured environments.

2.  Explore performance on other real-world problems or synthetic graphs with varying structural properties.

3. In Figure 1, the horizon $H=3$ appears to match the number of states. Is this property universal for DAMDPs? From Definition 2.2, the horizon $H$ seems always equal to the number of layers in an LDAG. Clarify whether this holds for all DAMDPs and, if not, provide examples where this differs.

4. Section 3.1 is difficult to follow without an example. Use a detailed scenario, such as wind farm optimization, to illustrate:
     - How atomic transitions and rewards relate to full state and action dynamics.
     -  The benefits of DAMDP over standard MDPs.

5. Provide a detailed analysis of how computational costs scale with the number of equivalence classes and graph size. How do equivalence classes and graph size affect efficiency? How does DAMDP compare to standard MDPs with similar state/action space sizes?

**Strengths And Weaknesses:**

**Strengths**:

1. The DAMDP framework extends MDPs with DAG-based structures, enabling modeling of real-world problems like wind farm optimization and network flow.
2. The paper provides a rigorous analysis of Bayesian regret for PSGRL, showing clear efficiency gains.

3. Real-world applications (e.g., wind farms) and experiments provide compelling evidence of the framework’s applicability.

 4. Definitions and explanations (e.g., state/action decomposition, equivalence classes) are precise, though hard to grasp at first glance.

**Weaknesses**:

1.   The assumption of known DAG structures may limit applicability to cases where such structures must be inferred.

2.   While leveraging DAG structures reduces complexity, the scalability for very large graphs with many equivalence classes is unclear.

3.  Comparisons primarily focus on PSRL. Evaluating against other RL methods (e.g., model-based RL) would strengthen empirical claims.

4.   The framework focuses on discrete state/action spaces. It is unclear whether we could extend to continuous spaces, as mentioned in future directions, which would broaden applicability.

---

> ### Author Response · Authors · 2024-12-17
> **Reply**
>
> Many thanks for your feedback and enthusiasm about our paper. We are grateful that you recognise that we provide a “rigorous analysis” and that our “explanations are precise”.
>
> We recognise that there is a lot to assimilate in the paper, especially notation-wise. We propose adding a table with all the notation used in the appendix. Hopefully, this will simplify the reading experience.
>
> We now address the weakness you listed:
>
> 1. You are correct that our method requires the structure to be known. It would be of great interest to infer the structure as we interact with the environment. While we recognise this as an interesting open problem, it is beyond the scope of the current paper, which aims to understand how RL algorithms can leverage known underlying graphical structures. The main challenge in learning the graph from interaction data is that it includes a search over a potentially very large (exponential in the number of nodes) class of potential DAMDPs. Our work is an important first step in this direction, as it provides a posterior sampling-based algorithm that successfully leverages graphical structures and an upper bound on the Bayesian regret that guarantees the proposed algorithm outperforms algorithms that ignore this structure.
>
> 2. We agree that the computational analysis of our algorithm is missing, and it would be of great interest to better understand how the algorithm's computational complexity is affected by the graphical structure. We will add this analysis to the paper; thanks for the suggestion!
>
> 3. Our experiments compare the proposed algorithm, PSGRL, with PSRL to validate our theoretical findings. Our experiments support our findings that a posterior sampling-based algorithm that can leverage the underlying structure outperforms a posterior sampling-based algorithm that ignores this latent structure.
>
> 4. The graphical structure allows for a decomposition of conditionally independent state variables. This benefit could be leveraged in both continuous and discrete settings. Designing an algorithm with the same principle in a continuous setting is plausible and interesting but is left for future work.
>
> We now address the requested changes:
>
> 1. We maintain that PSRL is a very natural baseline to evaluate the performance of PSGRL. Both algorithms rely on similar mechanics, i.e. sample a problem instance at random from a posterior distribution and use a planning algorithm to solve it. The main difference between the two algorithms is the construction of the posterior distributions from which they sample. If there is an additional natural baseline we have overlooked, we would be glad to take suggestions of additional baselines to validate our theoretical findings.
>
> 2. We provided a synthetic environment with the maximum leaky flow problem, for which we can create many problem instances (with different latent graphical structures). The paper evaluates the proposed algorithm in that environment with seven different latent graphical structures. In addition to this synthetic environment, we proposed to use the algorithm to optimise the yield of a wind farm. We built an environment based on a wind farm simulator and showed that our algorithm was more efficient than PSRL in finding the optimal yaw setting in two different wind farm layouts.
>
> 3. Yes, the horizon corresponds to the number of layers in the LDAG; this is specified in Section 3, and we will also emphasise this in Figure 1.
>
> 4. We propose to present the small abstract problem depicted in Figure 1 as a simple instance of the leaky flow problem, which could be used as a companion example. This will help the reader grasp the problem quickly; thanks for the suggestion.
>
> 5. We will provide an analysis of the computational cost of the proposed algorithm in section 5. Thanks for the suggestion.
>
> We hope that the above clarifications will reinforce your enthusiasm about our work.
>
> Sincerely,
>
> The authors

---

> > ### Comment · Reviewer_yEM6 · 2025-02-07
> > **Thanks for the reply!**
> >
> > Thanks for your response! I understand that inferring unknown DAG structures is particularly challenging in the initial steps.
> >
> > I appreciate the additional analysis of computational complexity—it has significantly improved my understanding of the method.
> >
> > Many of the newly added details in the figures and introduction have effectively addressed my concerns.
> >
> > Overall, I believe the authors have sufficiently addressed my feedback.

---

### Review · Reviewer_xpnV · 2024-12-07

**Summary Of Contributions:**

In a tabular finite-horizon MDP, the paper proposes to structure the state-action space with a “layered directed acyclic” graph, thus defining DAMDPs. The main idea is that states are now a combination of “atomic states” which are the nodes of a graph. Given a full state, the transition to the next full state is then defined by the product over probabilities defined by the graph structure. This modification of the value function definition is shown to preserve the good properties of RL, namely dynamic programming, and the authors show they can naturally extend algorithms based on extended value iteration, such as PSRL (Osband et al., 2017).

**Audience:**

Yes

**Broader Impact Concerns:**

Nine

**Claims And Evidence:**

Yes

**Requested Changes:**

A nice and complete Figure 1 that pedagogically explains the proposed structure and the notation

**Strengths And Weaknesses:**

## Strengths:
* Clear contribution: Show that a DAG structure can be used in combination with an MDP
* Well-written: the paper is well structured and well written overall, see comments below
* Experiments are nicely chosen to illustrate the benefits and limitations of the method

## Weaknesses:
* Perhaps limited impact? It is hard to judge but I guess so far people used function approximation (linear first, and then more general) to model the structure, and I am not sure why the graph would be more powerful
* Section 3 would greatly benefit a more complete figure to illustrate the setting

## General comment:

1/ I believe this is an interesting proposal which can have some practical applications and bridge the tabular setting and the function approximation setting more commonly used when the state-action space is structured. Perhaps this connection could be made more explicitly?

2/ Overall, the paper is well written and the contributions are clear. I believe that the notation is heavy and could deserve a bit more pedagogical presentation, typically via a Figure more complete than Figure 1, that should actually show what are “contexts”, and the related transition probability definitions. It took me a while to understand what are the different elements of your structure and Figure 1 only partially helps. The “All contexts” illustration is actually confusing, the “atomic action” y is either one arrow or 2 arrows? Why is it the same notation? I don’t understand the second column very well, what are the orange nodes? Anyway, what I mean is that in Section 3, you introduce a lot of notation and concepts like the equivalence classes, and this Figure does not help us processing them. I think it’d be really helpful if it would.

3/ Unless I missed something, you do not comment on the discounted MDP setting. Is the finite-horizon crucial in preserving the dynamic programming consistency? Also, your transitions and rewards don’t explicitly depend on t, except through the graph structure, but you chose to denote U_x the  number of parameters for each component of the MDP instead of H. Can you say how H compares to U_x, for x\in{r,\tau}?

## Minor comments:
A/ In Section 5, there’s a repeated sentence:  “We observe that there are 6 nodes with a single incoming edge (i.e. m1 τ = 6) and 9 with two incoming edges (i.e. m2 τ = 9)” is said almost exactly identically a couple of lines above.

B/ The line graph on the left of Figure 2 seems to suggest that DAMDPs are not so useful for these types of structures, and on Figure 3, the wind turbines you consider seem to be precisely organized as line graphs. I see that the wind turbines are in a grid, not just a line, but perhaps this choice of structure (and why it’s relevant) could be better discussed?

---

> ### Author Response · Authors · 2024-12-17
> **Reply**
>
> Thank you. We appreciate the time and effort you put into reviewing our work.
>
> Regarding your comment on the impact of our contribution, we highlight that our objective here is to gain a deeper understanding of the learning dynamics involved when the problem has a known underlying graphical structure. Our contributions, i.e. a new posterior sampling-based algorithm that can leverage this underlying structure and an upper bound on the suffered Bayesian regret, make a significant step toward understanding the sources of the observed efficiency gains (i.e. leveraging the repetitive atomic structure in the graph). It would be interesting to better understand how the proposed framework could work with function approximation. An immediate way to use this framework with function approximation would be to consider a unique set of parameters for each node’s class.
>
> In what follows, we address your general comments:
>
> 1. As we mentioned above, extensions with function approximations are interesting. However, we emphasise that our framework is not intended to replace the function approximation setting, but instead could be combined with function approximation to achieve further efficiency gains. We will discuss it as a future work direction. Still, we recall that our objective in this paper was to better understand the efficiency gain when a latent graphical structure is present.
>
> 2. Thank you for your detailed feedback on Figure 1. We agree that it would be beneficial if it could illustrate the equivalence classes and the varying action spaces.
> The previous figure consisted of three schemas; the updated figure (available in the new draft) contains six schemas. We now illustrate the graph, the state space, the action space, the reward context, the transition context, and finally, the time steps.
> For clarity, we recall that our construction allows for two contexts per equivalence class: the transition context and the reward context. These represent the atomic information required to compute the atomic transition and atomic reward function, respectively. Regarding the action space, we want the framework to be as general as possible. We would like to work with nodes with varying numbers of edges, i.e., different atomic action spaces. This is why, in Figure 1, we present two atomic action spaces, one with a single edge and one with two edges. We have made this more explicit in Figure 1.
>
> 3. The underlying graphical structure we consider is tightly linked to the horizon, i.e. H is the depth of the DAG, G. In that context, considering a discounted setting would be interesting (i.e. the graph would have an infinite depth), but we believe it lies outside our work's scope, as it will require tools to work with potentially infinite graphs. We will, however, mention this as an interesting direction for future work.
>
> Thank you also for your minor comments; we will amend those changes in our draft.
>
> We hope the above clarifications dissipate your concerns about our work and reinforce your belief that the present work would be valuable for the community.
>
> Sincerely,
>
> The authors

---

> > ### Comment · Reviewer_xpnV · 2025-01-13
> > **Figure 1 became more confusing**
> >
> > Dear authors,
> >
> > The new Figure 1 now has these "y" atomic action but we don't know what it is really, one is big, one is small, is the big one supposed to cover the 2 arrows? Re-reading Section 3 after a few weeks I realize it's still very hard to follow. I like reviewer yEW6's suggestion to use a concrete example to introduce your notation and concepts. I feel like you dismissed it a bit too quickly because Figure 1 is not a "small abstract problem", it is several small abstract situation and mini graphs that do not clearly illustrate your notation. The last graphs on the right, with the transitions, just does not show anything, or at least I don't see what it tries to show.
> >
> > The concept of "atomic" states and actions should be better distinguished from regular states and actions as people know it. The idea of using the wind farm as a running example is good since it's the typical structure you are trying to capture. It does ask for a bit of rewriting but I think it would really make the paper more readable, especially if making a clear Figure 1 is so hard.

---

> > > ### Author Response · Authors · 2025-02-03
> > > **Figure 1 and companion examples**
> > >
> > > Many thanks for your additional feedback.
> > >
> > > Regarding your feedback on Figure 1, we want to highlight that it shows a single abstract setting and each box depicts a component of the setup. We further modified Figure 1 to improve its clarity by adding a description of which graph quantity determines which aspects of the DAMDP. We agree that the different font size for “y” might be a source of confusion. In this new version the font size is standardized in all diagrams.
> > >
> > > We underscore the fact that the dimensionality of the atomic action space might depend on the number of outgoing edges. This is illustrated in the first box of the second row. We can see two nodes, one with a single outgoing edge whose atomic action lies in atomic action space $\mathcal{Y}_1$, and below we see another node with two outgoing edges whose atomic action lie in the atomic action space $\mathcal{Y}_2$. In many examples, e.g. the leaky graph flow, we may need to pick one ‘action’ per outgoing edge, so $\mathcal{Y}_2$ could have larger dimension than $\mathcal{Y}_1$. Detailed explanations are also provided in the caption of Figure 1.
> > >
> > > The last box of Figure 1 shows the two different transition contexts. We explain in the caption and in Section 3 that there are different atomic transition functions depending on the number of incoming edges. The two graphs with the orange nodes and edges (to depict the transition context, i.e. the parent nodes’ atomic state and actions) show the two transition functions present in our simple DAMDP.
> > >
> > >
> > > We agree with you that reviewer yEW6’s suggestion is a great one and in our reply to them we clearly indicated our intention to include a companion example in our final draft.
> > > To illustrate this, we upload a new version of the paper that includes two examples at the beginning of Section 3. One unusual aspect of our model is that we allow different nodes to have different atomic action spaces. To clarify this, we show how to model a wind farm as a DAMDP and how to model the leaky maximum flow problem as a DAMDP. The wind farm optimisation task is a particular DAMDP where the atomic action space remains the same for all nodes in the graph. In this context, the atomic actions control the yaw angle for a single wind turbine. Alternatively, to model the maximum leaky flow problem as a DAMDP, we require the atomic action space to depend on the number of outgoing edges. Here, the atomic action specifies how the flow should be distributed over all the available pipes (outgoing edges). Hopefully, these examples will clarify the setup and the concept introduced in section 3.
> > >
> > >
> > > Sincerely,
> > >
> > > The authors

---

### Review · Reviewer_Djn6 · 2024-12-07

**Summary Of Contributions:**

This work proposes a new subclass of the general MDP, i.e., DAMDP, which associates the state-action space with graphical structures. This subclass may be able to model some practical problems (e.g., the wind farm example provided by the authors). Theoretically, the newly introduced graphical structure allows algorithm design to speedup exploration and learning. As concretely demonstrated by the proposed design, a Bayesian regret bound can be achieved which is better than the lower bound in the general setting. Experiments are reported in two examples to verify the performance of the proposed algorithm.

**Audience:**

Yes

**Broader Impact Concerns:**

This work is mainly theoretical and considering the extension of current RL studies, which I do not foresee major negative social impact.

**Claims And Evidence:**

Yes

**Requested Changes:**

I would encourage the authors to think about the points mentioned in the above weakness discussions, especially how to highlight the difference/contribution of the algorithmic design and analyses of this work.

**Strengths And Weaknesses:**

Strength:
- It is an interesting and potentially practically consideration to involve graphical structures to capture the state-action spaces in MDP. This may inspire follow-up research towards a closer connection between RL and graph learning.

- The overall writing is clear (although some modifications would be nice, as detailed in the following) and the authors did a good work in structuring the work and highlighting the contributions. The graphical illustrations are very helpful.

- Rigorous theoretical results are provided for the proposed algorithm. In particular, the illustration that it can achieve better performance than the general lower bound is important to highlight the effectiveness of leveraging the structures.

- The experiments are nice practical verifications of the algorithmic effectiveness.

Weakness
- The discussions on the planning, i.e., Section 3.2, seems straightforward to me without much contributions. Especially, given that the DAMDP is a subclass of MDP, many results seems as standard corollaries and extensions from MDP. If I missed anything specific to DAMDP, let me know.

- The algorithmic design could be elaborated better. Especially, I believe that the key difference is how to leverage the graphical structure, which is however largely omitted in the main paper. Without that, the algorithm seems just as a common posterior sampling algorithm. I would encourage the authors to involve more discussions on how the algorithm leverage the structure.

- Similarly, for the theoretical analyses, I believe it would be important to further highlight the uniqueness/contribution done in this work, which I believe lies in the construction of confidence sets, while that part is currently not discussed in the main paper.

- Even with the specific construction of confidence sets, I still feel that the analyses largely follow previous works. In that sense, to highlight the problem class studied in this work, it would be more desired to show that different previous algorithms, not just PSRL, can be extended effectively with better performances, e.g., those in non-Bayesian setting and the model-free ones.

---

> ### Author Response · Authors · 2024-12-17
> **Reply**
>
> Many thanks for your comments and thoughtful feedback. We are glad that you find our work “interesting”, “practical”, and “inspiring”.
>
> We agree that the planning analysis follows from DAMDP’s properties inherited from the underlying MDP. Nevertheless, we use the atomic operator to build the corresponding planning algorithm. Consequently, this differs from standard planning algorithms, so it makes sense to provide the appropriate guarantee that it works as expected. Additionally, as the planning algorithm is an important building block of the resulting posterior sampling algorithm, it seems reasonable to describe it in the main paper.
>
> The precise description of how the algorithm leverages the graphical structure to boost its efficiency is given in the proof, which is deferred to the appendix. We understand that this limits the reader's ability to appreciate the novelty of our approach and the graph structure's impact on the algorithm efficiency from reading only the main paper. Therefore, we will add a sub-section in section 5, which describes the algorithm and the construction of the confidence set. Hopefully, this will illustrate better how the graphical structure is leveraged. Thanks for the suggestion!
>
> As you point out, the confidence sets we construct are different to those considered in prior work. We will add more details about their novelty to the main text. Note that the construction of these confidence sets would in fact allow us to also construct an optimistic algorithm for DAMDPs. Indeed, similar to UCRL2, this optimistic algorithm would select a policy by solving the bilevel optimisation problem $max_{\pi \in Pi}\max_{M \in \mathcal{M}_G^k} V^\pi_1(s; M)$ for each full state $s$ in each episode $k$. Solving this optimisation problem in every episode will be computationally challenging since there are dependencies on the transitions between different layers of the graph hidden in $\mathcal{M}_G^k$, which means that the actions taken at different nodes in the same layer are not independent (note that this also means that the results from Neu & Pike-Burke (2020) cannot be applied). In contrast, our posterior sampling-based algorithm avoids solving such an optimisation problem as we can simply use our planner to jointly select all the atomic actions of a given layer in the sampled DAMDP.
> We will discuss this potential extension in the appendix to highlight that our framework is not limited to posterior sampling algorithms. However, since Thompson sampling-based algorithms are computationally more efficient, they remain our main focus in this paper.
>
>
> We hope our clarifications reinforce your belief that our paper is valuable to the community.
>
> Sincerely,
>
> The authors

---

### Author Response · Authors · 2024-12-17
**Reply to all reviewers**

We are grateful for the time and effort the reviewers dedicated to our work, which will strengthen our article. We are also pleased to see that the reviewers were enthusiastic about our work, especially on the contributions: “interesting”, “may inspire follow-up research”, “interesting proposal”, and on the writing: “very clear and the paper is well structured”, “details are well explained”, “definitions and explanations are precise”.

Please find a response for each review question as a direct reply to the reviewer asking it.
In our response, we mention the changes we plan to make in the paper; if you agree with the proposed modifications, we will modify the paper accordingly. For now, the latest version of the uploaded paper includes a revised version of Figure 1 and a table for the notation.


Sincerely,

The authors

---

### Decision · Action_Editor_jGmm · 2025-02-20

**Recommendation:** Accept with minor revision

**Comment:**

Given that the claims are clearly supported by evidence, that there is interest in the TMLR community, the writing is clear, and rigorous theoretical results are provided for their algorithm, I recommend acceptance.

Please finish all changes you described in your "reply to all reviewers".

**Audience:**

Yes, there is an audience for this; though it seems small.

**Claims And Evidence:**

The reviewers were unambiguous that the claims made by the paper were accurate, convincing, and clear.